# Emissions of Carbonaceous Particulate Matter and Ultrafine Particles from Vehicles—A Scientific Review in a Cross-Cutting Context of Air Pollution and Climate Change

Bertrand Bessagnet [1,*], Nadine Allemand [2], Jean-Philippe Putaud [1], Florian Couvidat [3], Jean-Marc André [2], David Simpson [4,5], Enrico Pisoni [1], Benjamin N. Murphy [6] and Philippe Thunis [1]

[1] Joint Research Centre, European Commission, 21027 Ispra, Italy; jean-philippe.putaud@ec.europa.eu (J.-P.P.); enrico.pisoni@ec.europa.eu (E.P.); philippe.thunis@ec.europa.eu (P.T.)
[2] Citepa, 42 Rue de Paradis, 75010 Paris, France; nadine.allemand@citepa.org (N.A.); jean-marc.andre@citepa.org (J.-M.A.)
[3] INERIS, Parc Technologique Alata, BP 2, 60550 Verneuil-en-Halatte, France; florian.couvidat@ineris.fr
[4] EMEP MSC-W, Norwegian Meteorological Institute, 0313 Oslo, Norway; david.simpson@met.no
[5] Department Space, Earth & Environment, Chalmers University of Technology, SE-412 96 Gothenburg, Sweden
[6] Office of Research and Development, U.S. Environmental Protection Agency, Research Triangle Park, Durham, NC 27711, USA; murphy.ben@epa.gov
[*] Correspondence: bertrand.bessagnet@ec.europa.eu or bertrand.bessagnet@lmd.ipsl.fr

**Featured Application: Key conclusions and recommendations are proposed to enlighten decision makers in view of the next regulations on vehicle emissions in Europe and worldwide through the synergistic contexts of air quality and climate change.**

**Abstract:** Airborne particulate matter (PM) is a pollutant of concern not only because of its adverse effects on human health but also on visibility and the radiative budget of the atmosphere. PM can be considered as a sum of solid/liquid species covering a wide range of particle sizes with diverse chemical composition. Organic aerosols may be emitted (primary organic aerosols, POA), or formed in the atmosphere following reaction of volatile organic compounds (secondary organic aerosols, SOA), but some of these compounds may partition between the gas and aerosol phases depending upon ambient conditions. This review focuses on carbonaceous PM and gaseous precursors emitted by road traffic, including ultrafine particles (UFP) and polycyclic aromatic hydrocarbons (PAHs) that are clearly linked to the evolution and formation of carbonaceous species. Clearly, the solid fraction of PM has been reduced during the last two decades, with the implementation of after-treatment systems abating approximately 99% of primary solid particle mass concentrations. However, the role of brown carbon and its radiative effect on climate and the generation of ultrafine particles by nucleation of organic vapour during the dilution of the exhaust remain unclear phenomena and will need further investigation. The increasing role of gasoline vehicles on carbonaceous particle emissions and formation is also highlighted, particularly through the chemical and thermodynamic evolution of organic gases and their propensity to produce particles. The remaining carbon-containing particles from brakes, tyres and road wear will still be a problem even in a future of full electrification of the vehicle fleet. Some key conclusions and recommendations are also proposed to support the decision makers in view of the next regulations on vehicle emissions worldwide.

**Keywords:** emissions; SVOC; IVOC; organics; PAH; black carbon; brown carbon; air quality; vehicles; climate

## 1. Introduction

Ambient air pollution causes significant excess mortality and loss of life expectancy (LLE), especially through cardiovascular diseases. Ambient air pollution causes LLE similar to that of tobacco smoking. The global mean LLE from air pollution (2.9 years) exceeds that

by all forms of violence (0.3 years), by one order of magnitude based on calculations of exposure to Ozone and PM$_{2.5}$ [1]. The need to reduce emissions, improve air quality and reduce the impacts on public health and the environment on the one hand, and questions of cost, technical feasibility and societal acceptability on the other hand are key issues [2].

Airborne particulate matter (PM) is a pollutant of concern not only because of its adverse effects on human health, but also on visibility, soil quality, buildings, infrastructure, and climate [3–5]. PM consists of particles with a very wide range of sizes and chemical compositions, with black carbon (BC) and organic carbon (OC) as key components. According to Southerland et al. [6], regional averages of PM$_{2.5}$-attributable deaths increased in all regions except for Europe and the Americas, driven by changes in population numbers, age structures, and disease rates. In 2021, the World Health Organisation (WHO) proposed more stringent guidelines for PM$_{2.5}$ (5 $\mu$g m$^{-3}$ instead of the previous guideline of 10 $\mu$g m$^{-3}$) and recommended "good practices" highlighting the role of BC and ultrafine particles (UFP), and encouraging national authorities to better monitor BC and UFP concentrations and take actions [7]. BC was recognised in cohort studies of French adults to partly explain the association between PM$_{2.5}$ and lung cancer [8]. Among the main toxicological mechanisms associated with UFPs is the oxidative stress generated from reactive oxygen species associated with constituents such as metals, polycyclic aromatic hydrocarbons (PAHs), and BC. The main effects of UFP on health include respiratory problems such as asthma and chronic obstructive pulmonary disease, pulmonary fibrosis, neurodegenerative diseases, cardiovascular diseases, DNA changes generated from epigenetic changes, and genotoxic, mutagenic and carcinogenic activity [9].

BC plays a key role in the Earth's climate system because it absorbs solar radiation, influences cloud processes, and alters the snowmelt and ice cover. In contrast, OC has an overall cooling effect due to its scattering optical properties, although brown carbon (BrC) from organic matter (OM) could have the opposite effect. BC concentrations respond quickly to reductions in emissions because BC is rapidly removed from the atmosphere by deposition. Thus, BC emission reductions represent a potential mitigation strategy that can reduce global climate forcing from anthropogenic activities in the short term and slow the associated rate of climate change even if in the last Intergovernmental Panel on Climate Change (IPCC) suggest that the effect of BC could be lower [10]. Carbonaceous emissions also have an impact on meteorology at the local scale, particularly in mountainous regions, through a complex suite of processes involving the atmospheric water column [11].

Road traffic is the major air pollution source in urban areas [12–15] due to fuel combustion (exhaust-traffic-related emissions), and brake/tyre wear (non-exhaust-traffic-related emissions). This urban contribution on PM concentrations varies greatly from region to region, with the current highest contribution from Africa and India and the lowest from Europe and North America due to the application of stringent standards for vehicle emissions [16]. The restrictions implemented during the COVID-19 pandemic provided an "opportunity" to show the major role of traffic emissions on PM and UFP concentrations in highly urbanised areas [17]. However, it has been demonstrated that diesel vehicles are responsible for approximately 75% of the total costs of air pollution related to road transport in Europe [18]. Road traffic is also recognised as the second-largest contributor of BC emissions (26%) in the European Union (EU), with major importance in cities [19]. Recent work by Caubel et al. [20] in the US showed the high spatial variability of BC concentrations in Oakland (California) using a dense network of low-cost sensors, highlighting the exposure of the population to vehicle emission. Stringent regulations on vehicle emissions was shown to reduce BC and OC concentrations in a study on urban sites in Canada in the period 2003–2019 [21], although the decrease in OC is less pronounced and probably counterbalanced by secondary organic aerosol (SOA) formation.

The goal of this review is to compile and analyse the recent literature on emissions of carbonaceous PM and its precursors assigned to the road traffic sector. Particle number and polycyclic aromatic hydrocarbons (as OM) are species closely linked to carbon issues and will be part of this review. Some key conclusions and recommendations are proposed to

enlighten decision makers in view of the next regulations on vehicle emissions in Europe and worldwide through the synergistic contexts of air quality and climate change.

## 2. General Context

### 2.1. Policy Context

The Convention on Long-Range Transboundary Air Pollution (CLRTAP) of the United Nation Economic Commission for Europe (UNECE) signed in 1979, and entered in force in 1983, was the first international instrument aiming at reducing air pollutant emissions and decreasing their impacts on health and ecosystems. Several protocols have been implemented since and the latest, the Protocol to abate Acidification, Eutrophication and Ground-level Ozone or the Gothenburg Protocol amended in 2012, entered into force on 7 October 2019.

The objective of the Amended Gothenburg Protocol (AGP) is to control and reduce emissions of sulphur, nitrogen oxides, ammonia, volatile organic compounds and particulate matter that are caused by anthropogenic activities. These emissions are indeed expected to cause adverse effects on human health and the environment, natural ecosystems, materials, crops and the climate in the short and long term, due to acidification, eutrophication, particulate matter or ground-level ozone as a result of long-range transboundary atmospheric transport. The protocol also aims at ensuring that atmospheric depositions or concentrations do not exceed the ceilings over target regions in the long term through a stepwise approach that accounts for advances in scientific knowledge. Updating and assessing on a regular basis the information on emission abatement technologies for the reductions in the atmospheric emissions of $SO_2$, NOx, volatile organic compounds (VOCs), PM (including $PM_{10}$, $PM_{2.5}$ and BC), heavy metals and persistent organic pollutants (POPs) from stationary and mobile sources, including the costs of these technologies, are important tasks within the CLRTAP [22].

A new report [23] issued by the Joint Research Centre (JRC) and the Organisation for Economic Co-operation and Development (OECD) calls for ambitious policy action to reduce air pollution in Arctic Council countries (Canada, Denmark, Finland, Iceland, Norway, the Russian Federation, Sweden, and the United States), highlighting the environmental, health, and economic benefits of air quality improvements. The Arctic environment is particularly sensitive to short-lived climate pollutants, due to their strong warming effect. In particular, BC, which is the most light-absorbing component of particulate matter ($PM_{2.5}$), not only contributes to the negative impacts of air pollution on human health, but is also a major contributor to Arctic warming. In 2021, the Arctic Council countries have established policy actions to reduce their BC emissions, setting a collective 2025 target of lowering BC emissions by 25–33% from 2013 levels [24].

In the EU, the National Emission reduction Commitments Directive [25] is the general framework which transposes AGP obligations to the EU legislative level, defining for instance the national emission reductions for the main pollutants among a long list of measures to combat air pollution. National reduction commitments for PM emissions are defined for $PM_{2.5}$. However, BC is mentioned without binding decisions but with recommendations to measure this substance both at the emission and in ambient air. It is also encouraged to take synergic measures reducing both PM and BC to both (i) curb climate change effects and (ii) improve air quality. PAHs are trace carbonaceous compounds with potential carcinogenic effects. While PAHs are not cited in the AGP, they are addressed by the amended Aarhus Protocol on POPs [26].

The Air Quality Directive [27] defines objectives for ambient air concentrations in the EU, designed to avoid, prevent or reduce harmful effects on human health and the environment as a whole. To this end, it sets out measures for the assessment of ambient air quality in Member States as well as for obtaining information on ambient air quality in order to help combat air pollution and nuisance. The Directive also aims at increasing cooperation between the Member States in reducing air pollution. For $PM_{10}$, there are two limit values set to 40 and 50 µg m$^{-3}$—annual and daily bases. PAH concentrations

are also regulated [28]. In contrast, no limit values are set for BC or OC, but there is an obligation to measure these substances in rural areas to ensure that adequate information is available on $PM_{2.5}$ concentration and composition in the background. This information is essential to judge the enhanced levels in more polluted areas (such as urban background, industry related locations and traffic-related locations), assess the possible contribution from long-range transport of air pollutants, support source apportionment analysis and for use in modelling in urban areas to further understand specific pollutants such as particulate matter. So far, no limit values have been set for PN concentrations in ambient air.

Road traffic remains a major emitter of primary carbonaceous species [29] although, at a global scale, primary PM anthropogenic emissions are mainly driven by residential combustion [30] and wildfire (Table 1).

**Table 1.** Sectoral emissions of particulate matter in 2010, ECLIPSE V5a, Gg year$^{-1}$ [30]. (a) Values are middle-of-the-range estimates based on the ranges reported in [31,32], and based on global fuel consumption and ranges of emission factors [33]. (b) GFED3.1 [34,35] without agricultural waste burning that is included based on GAINS [36] estimates in category "Agriculture"; $PM_{10}$ value based on TPM (total particulate matter); $PM_1$ not available in GFED—here assumed equal to $PM_{2.5}$.

| | $PM_{10}$ | $PM_{2.5}$ | $PM_1$ | BC | OC | OM |
|---|---|---|---|---|---|---|
| *Agriculture* | 6555 | 3848 | 2883 | 337 | 1313 | 2364 |
| *Residential combustion* | 23,078 | 21,857 | 20,742 | 4163 | 8852 | 15,329 |
| *Industrial processes* | 12,162 | 8340 | 4135 | 462 | 633 | 823 |
| *Large-scale combustion* | 11,561 | 6420 | 3812 | 136 | 164 | 248 |
| *Oil and gas, mining* | 1706 | 571 | 412 | 226 | 93 | 120 |
| *Transport—road* | 3339 | 2925 | 2524 | 1349 | 1116 | 1451 |
| *Transport—non-road* | 861 | 823 | 795 | 363 | 217 | 283 |
| *Waste* | 1388 | 1272 | 876 | 97 | 751 | 977 |
| *International shipping* | 1856 | 1758 | 1612 | 120 | 398 | 517 |
| *International aviation* [a] | 30 | 30 | 28 | 10 | 10 | 13 |
| *Global anthropogenic* | 62,537 | 47,843 | 37,819 | 7264 | 13,548 | 22,125 |
| *Forest and savannah fires* [b] | 48,207 | 33,014 | 33,014 | 2268 | 19,489 | 31,363 |
| *Global total* | 110,744 | 80,858 | 70,834 | 9532 | 33,037 | 53,489 |

In the EU, the so-called "Euro" regulations [37–39] are the legal framework to fix emission limit values (ELVs) for all types of vehicles. There are no specific standards on carbonaceous PM emissions in Euro regulations. However, carbonaceous species are a major constituent of PM particles emitted at vehicles exhaust (see Section 2.2). As summarised in Table 2, the last Euro 6 regulation includes ELVs for PM in mg km$^{-1}$ and also for particle numbers (PN) in # km$^{-1}$ for both passenger cars and light commercial vehicles [17]. BC emissions are among the core regulatory policy issues in all engine manufacturers, and they have not yet been adequately addressed by policy makers [40]. The way emissions are determined depends on the driving cycle used for emission measurements, as described in [41,42] for NOx emissions several years before the "Dieselgate" scandal [43]. For diesel vehicles, there was a clear divergence between real emissions and homologation limits values for NOx emissions.

Europe has taken a leading role in introducing particle number (PN) limits, effectively forcing the introduction of filters on diesel power train exhausts [44]. These provisions have been mirrored in the Chinese standards (Figure 1). The U.S. Tier 3 (US-LEV-III) standard, which will be phased in by 2025, sets a PM limit of approximately 1.9 mg km$^{-1}$ over the federal test procedure (FTP) cycle (Figure 1). Although no PN limit exists in the United States, California's LEV III standards set a PM limit of approximately 0.6 mg km$^{-1}$ over the same cycle, which is deemed comparable in stringency to Europe and China's PN limits. This limit will be fully phased in by 2028. Regarding heavy-duty vehicles (HDV), for instance the Chinese Ministry of Ecology and Environment (MEE) released in July 2018 the final rule for the China VI emission standard for HDV, hereafter referred to as China VI.

The China VI standard is among the world's most stringent HDV emission standards [45] and combines best practices from both European and U.S. regulations [46].

**Table 2.** Euro 6 emission limits for passenger cars and light commercial vehicles; table from Rodríguez et al. [44].

|  | LDVs, LCVs Class 1 [a] | | LCVs Class 2 | | LCVs Class 3 | |
|---|---|---|---|---|---|---|
|  | Gasoline [b] | Diesel [c] | Gasoline | Diesel | Gasoline | Diesel |
| NMHC * | 68 | - | 90 | - | 108 | - |
| THC * | 100 | - | 130 | - | 160 | - |
| NOx * | 60 | 80 | 75 | 105 | 82 | 125 |
| THC + NOx * | - | 170 | - | 195 | - | 215 |
| CO * | 1000 | 500 | 1810 | 630 | 2270 | 740 |
| PM * | 4.5 [d] | 4.5 [d] | 4.5 [d] | 4.5 | 4.5 [d] | 4.5 |
| PN ** | $6 \times 10^{11}$ [d] | $6 \times 10^{11}$ [d] | $6 \times 10^{11}$ [d] | $6 \times 10^{11}$ | $6 \times 10^{11}$ [d] | $6 \times 10^{11}$ |

Notes: (a) Classes 1 through 3 are weight classes. (b) Gasoline is used as a proxy term for positive-ignition (PI) engines. (c) Diesel is used as a proxy term for compression-ignition (CI) engines. (d) Applicable to direct-injection engines.* unit in mg km$^{-1}$, ** unit in # km$^{-1}$.

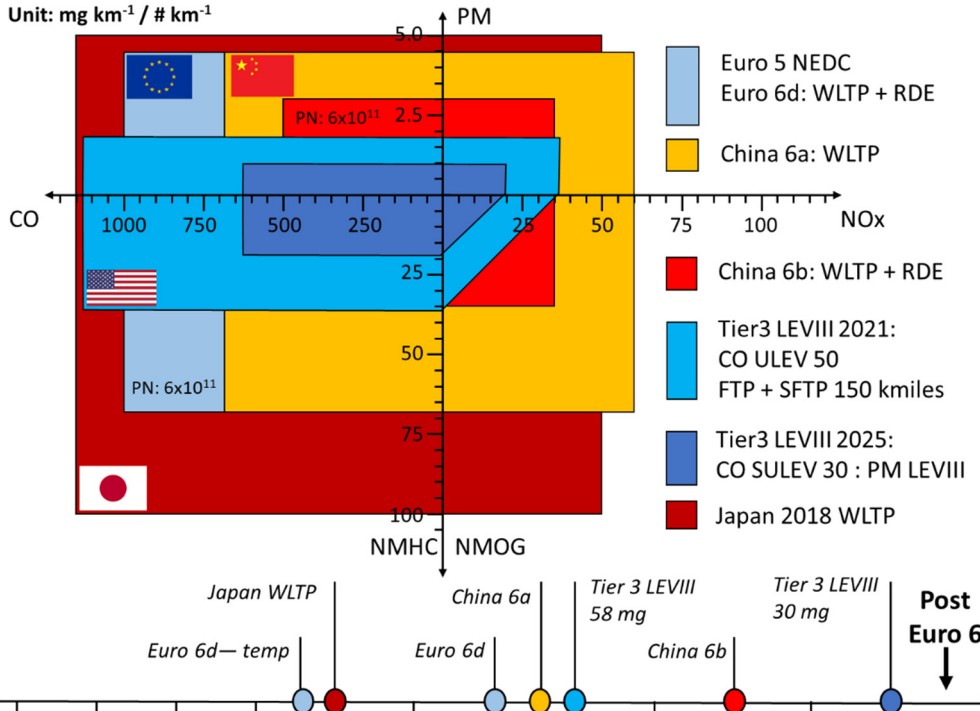

**Figure 1.** Simplified diagram of emission limits and phase-in timing in four of the main world regions (LEV: low—emission vehicle, ULEV: ultra-low-emission vehicle, WLTP: World LDV Test Procedure, FTP: Federal Test Procedure, SFTP: supplemental FTP, and RDE: real driving emissions).

## 2.2. Overview of BC/PAH/PM$_{2.5}$ Emissions

In the EU27+UK, the main emitting sector is the residential sector (Table 3), followed by road transport and open burning of waste (Figure 2). The lowest ratio BC/PM$_{2.5}$ (10%) reported in the EEA guidebook [30,47] is also for the residential combustion sectors (Figure 3) for solid fuels. This low ratio refers to previous works of [48–50] and is based on an emission factors measurement method which accounts for condensables (see Section 3.5). There are rather major issues surrounding condensable emissions for the residential sector in Europe [51]. Simpson et al. [52] summarises the issues in the context of national reporting under the CLRTAP and the EMEP. The concept of condensable is explained later in this review (see Section 3.5).

**Table 3.** Top ten BC-emitting activity sectors in EU28 in 2018 (sorted from the highest to the lowest emitter) according to the EMEP database [53].

| Code | Name |
|------|------|
| 1A4bi | Residential: stationary |
| 1A3bi | Road transport: passenger car |
| 5C2 | Open burning of waste |
| 11B | Forest fires (natural wildfires) |
| 1A3di(i) | International maritime shipping |
| 1A4cii | Agriculture/Forestry/Fishing: off-road vehicles and other machinery |
| 1A3bii | Road transport: light-duty vehicles |
| 1A3biii | Road transport: heavy-duty vehicles and buses |
| 1A2gviii | Other stationary combustion in manufacturing industries and construction |
| 1A2gvii | Off-road mobile sources in manufacturing industries and construction |

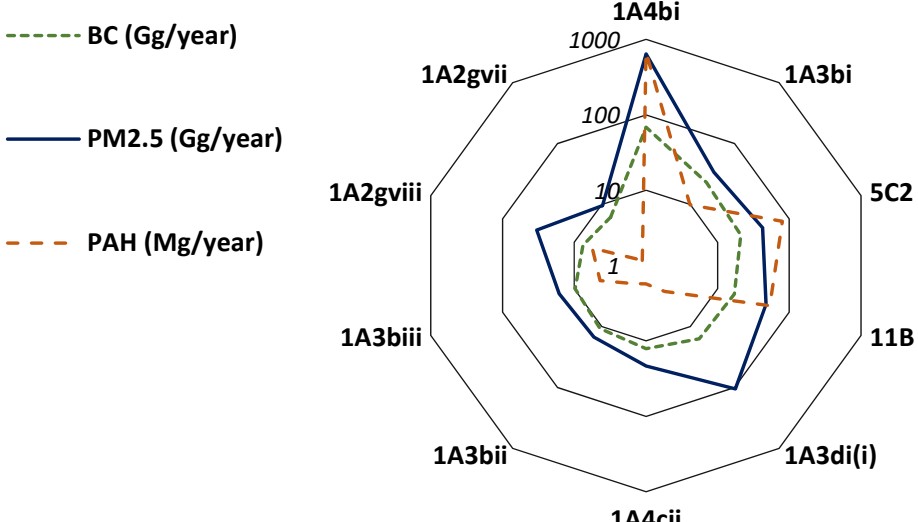

**Figure 2.** EU28 official emissions in kt year$^{-1}$ or t year$^{-1}$ (Gg or Mg year$^{-1}$) for BC, PM$_{2.5}$ and PAHs (as the sum of benzo(a)pyrene, benzo(b)fluoranthene, benzo(k)fluoranthene and indeno(1,2,3-cd)pyrene) in 2018 for the main emitting NFR sectors (refer to Table 3). Emissions are computed from the EMEP database [53].

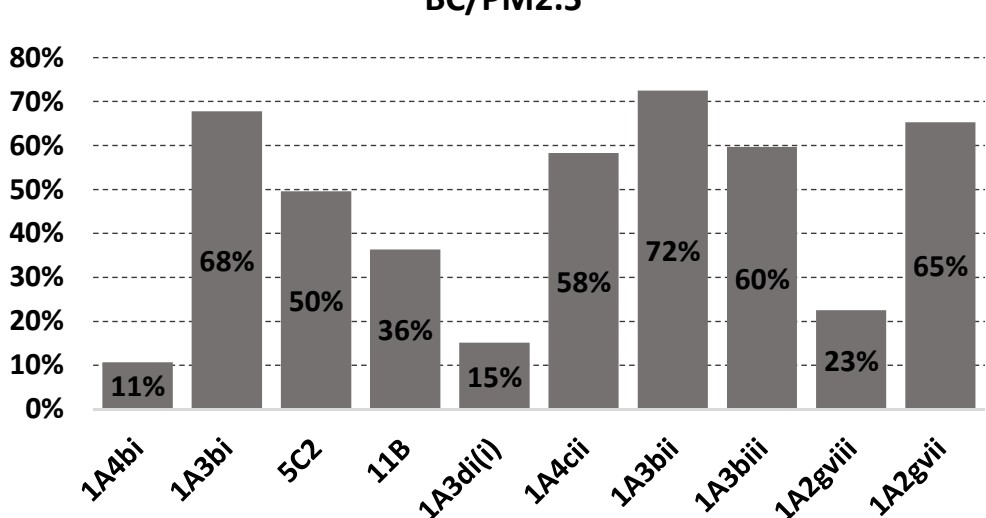

**Figure 3.** Ratio BC/PM$_{2.5}$ for EU28 official emissions in 2018 computed from the EMEP database [53] for the main emitting NFR sectors (refer to Table 3).

### 2.3. Air Pollution and Climate Change

OC concentrations result in an increase in scattering of solar radiation, and hence cool the climate system [4,54,55]. The magnitude and direction (warming or cooling) of biophysical climate effects vary and are strongly site specific. Due to its light absorption properties, BC has a direct warming effect of the atmosphere. When deposited on white surface, BC can also darken the snow/ice surface, affect the energy balance, and further lead to acceleration of the melting of the cryosphere (e.g., glaciers, snow cover, and sea ice). According to Kang et al. [56], the large BC-in-snow radiative forcing and associated snow albedo feedback lead to an acceleration in the total glacier melt (approximately 20%) and/or a reduction in the duration of the snow cover by several days, resulting in an increase in glacier discharge.

A "grey area" was highlighted in a recent communication [57] regarding the links between air pollution and climate change. There are numerous links between air pollution and climate change policies, but these are probably not very well integrated into international laws and there is little acknowledgement of potential synergies and trade-offs [58–60]. Recently, greater attention has been paid to short-lived climate pollutants (SLCPs), some of which contribute to both air pollution and global warming (e.g., BC and $CH_4$). Reducing emissions of BC is particularly important where SLCPs are concerned, but international law does not yet provide clear answers as to how such emissions are to be regulated. For BC, a recent study [61] showed that the major benefits in reducing BC emissions from China result from more stringent emission standards and the accelerated retirement of older heavy-duty diesel vehicles. The shorter atmospheric lifetime of BC compared to $CO_2$ implies that the mitigation of BC emissions would offer an important opportunity to alleviate global warming in the short term.

The Paris Agreement [62], which requires all countries to contribute to climate change mitigation, does not specify which greenhouse gases shall be tackled. The Paris Agreement is based on a bottom-up approach to mitigation, where individual countries define the action they will take and report on it through nationally determined contributions (NDCs), in which any greenhouse gas or substance can potentially be included. In fact, in their intended NDCs, many countries included methane, several mentioned SLCPs and some countries such as Mexico and Chile specifically mentioned BC. At the same time, the dominant focus of the UN climate change regime has been on reducing carbon dioxide emissions while less attention has been given to methane, and BC has hardly been discussed at all.

Through its 2012 amendments, the Gothenburg Protocol to the CLRTAP includes emission reduction targets for fine PM. Parties are encouraged to implement measures to achieve their national targets for particulate matter, to give priority, to the extent they consider appropriate, to emission reduction measures which also significantly reduce BC to provide benefits for human health and the environment and to help mitigation of near-term climate change (Section 2.2), and to report on their current BC emissions and projections.

Moreover, action on BC has been taken by the Arctic Council through the adoption of the Framework for Action on Enhanced BC and Methane Emission Reductions and, more recently, of a first collective regional goal for reducing BC emissions. However, the outputs of the Arctic Council are not legally binding. In sum, there is a clear gap in the regulation of BC emissions, as no legal frameworks of global reach are currently in place to cover this pollutant. The impacts of organic aerosol (OA) on climate change are rather complex, since they may contribute to cooling or warming (through the lensing effect of OA coating on BC cores, and "brown" carbon, see Sections 3.2 and 3.3), and there are many feedback mechanisms between the biosphere and climate change [63,64].

## 3. Scientific Background

### 3.1. Carbon: A Key Component of PM

Historically, most PM was emitted from coal burning and measured as black smoke [3,65]. However, in the second half of the 20th century in developed countries, there was a reduction in black smoke emissions from coal burning and PM steadily became dom-

inated by primary carbonaceous matter from road traffic and secondary pollutants, namely ammonium salts and secondary OC. Regarding the shape of fresh carbon particles emitted at the exhaust, a high-resolution transmission electron microscopy (HRTEM)analysis indicates that the fringe tortuosity and separation distance decrease as the fuel injection quantity increases, while the fringe length increases with increasing fuel injection quantity [66].

The particle size distribution of airborne PM is very broad, with particle diameters ranging from a few nanometres (newly formed particles) to tens of micrometres (e.g., crustal particles). The size of primary particles emitted by modern thermal engines ranges from approximately 10 to 100 nm [67]. There has been a great deal of interest in ultrafine (nano) particles because of suspicions of enhanced toxicity, and, as primary traffic particle emissions decreased, the frequency of particle nucleation events increased, contributing to particle number without affecting the total particle mass [3]. However, at the emission exhaust, the homogeneous nucleation [68] of organics is unlikely to occur, compared to heterogeneous processes involving organic vapours and pre-existing particles [69–71].

The composition of fine particles (referred to as $PM_{2.5}$) showing a contribution of carbonaceous species of 41%, 47% and 54% in London, Beijing, and Delhi, respectively, for wintertime campaigns [72] clearly shows the importance of carbonaceous compounds in megacities. In European background sites, Bressi et al. [73] showed a contribution of 35 to 65% of the OA in the fine fraction of particles (<1 μm). In a recent study in Colombia, the spatial distribution of eBC concentrations showed that vehicle emissions and traffic jams, as a consequence of road and transport infrastructure, are the factors that most affect eBC concentrations [74].

In general, internal combustion engine exhaust particles contain organic and elemental carbon (EC), metals and other elements [75,76]. In a recent study in China, Xiang et al. [77] demonstrated that BC from sources other than traffic have more and more importance even in urban areas. Carbon is the main element in particles emitted by diesel engines [78], as depicted in Figure 4. As the emission exhaust cools in the exhaust manifold and associated pipework, carbon particles agglomerate, forming high-surface-area material onto which uncombusted and partially combusted gaseous products are adsorbed, as well as sulphur oxides and nitrogen oxides (NOx) formed during high-temperature combustion in the cylinder. During the dilution of the exhaust aerosol to ambient conditions, a series of processes such as nucleation, coagulation ad- and absorption make particles, consisting of amorphous EC particle cores [79,80], more and more spherical with ageing (Figure 5), therefore lowering their fractal dimension [81–84]. Particulate OC also forms in the atmosphere through complex chemical pathways from primarily emitted gaseous organic compounds. This so called SOA species formation in urban areas was recently reviewed [85]. In U.S. cities, 20 to 60% of emissions, reactivity, and SOA-forming potential of anthropogenic VOCs are associated with mobile emission sources, demonstrating the impact of traffic on urban U.S. air quality [86,87]. The role of liquid phases in the particles has been demonstrated [88], using thermodynamic and kinetic modelling, so that the presence of three liquid phases in particles impacts their equilibration timescale with the surrounding gas phase. Three phases will likely also impact their ability to act as nuclei for liquid cloud droplets, the reactivity of these particles, and the mechanism of SOA formation and growth in the atmosphere.

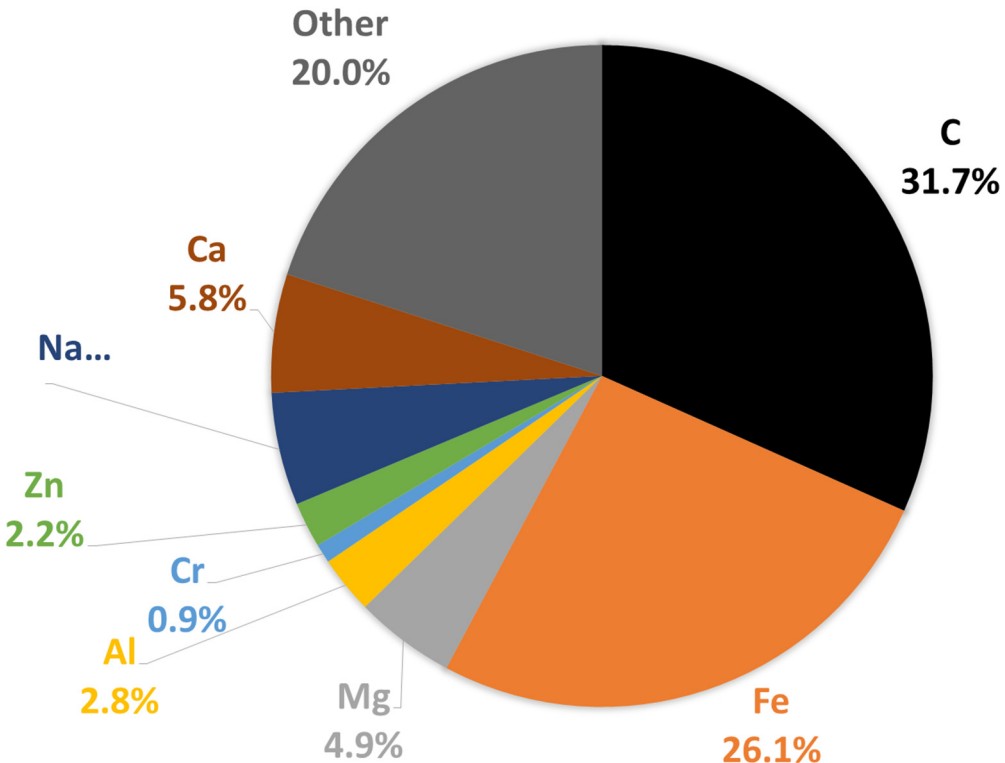

**Figure 4.** Typical elemental composition of diesel particles using laser-induced breakdown spectroscopy (LIBS). C: carbon, Fe: iron, Mg: magnesium, Al: aluminium, Zn: zinc, Na: sodium, and Ca: calcium (from Viskup et al. [78]).

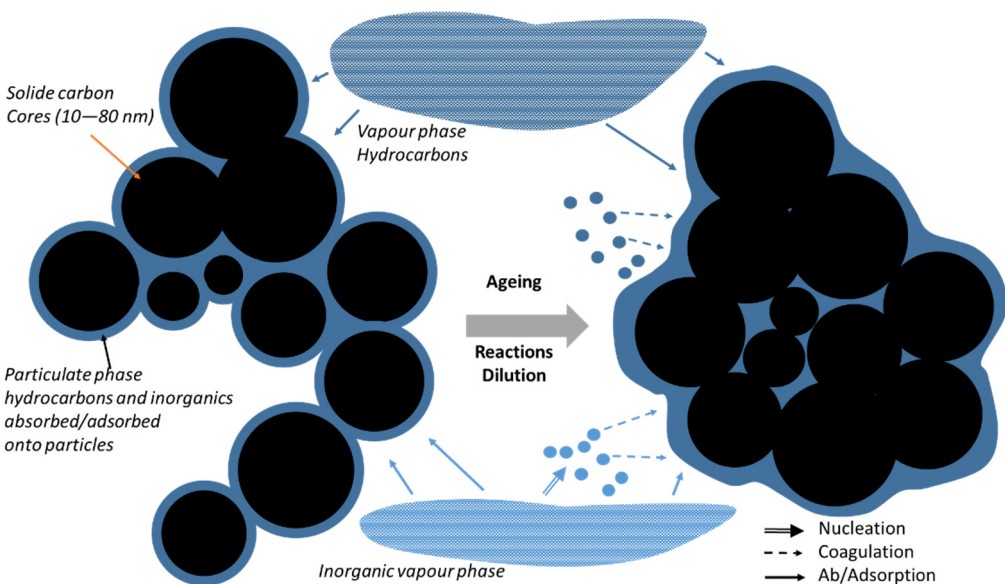

**Figure 5.** Schematic representation of diesel particulate matter (PM) formed during combustion of atomised fuel droplets and its evolution during the dilution until ambient conditions. The resulting carbon cores agglomerate and adsorb species from the gas phase.

The OM internal combustion engine exhausts also contain O, H, N, etc., atoms, and therefore OM/OC is mostly in the range 1.2–1.6 [89–91]. This ratio becomes higher as secondary oxidised species form and accumulate on primary exhaust particles.

Moreover, a recent study by Cao et al. [92] showed that the abundant transition metals detected on BC particles are possibly catalysers for the transformation from less to more oxidised OA in the aerosol aqueous phase and thus facilitate SOA formation. More studies

are needed to further explore the potential role of BC and transition metals as catalysers for OA aging and the co-benefit of BC and transition metals pollution control on SOA reduction in the atmosphere.

The evolution of the mixing state of particles, as depicted in Figure 5, is also important to better account the radiative effect of particles. Far from sources, particles are expected to be internally mixed and unfortunately most global models often work with the assumption of external mixing. This can lead to noticeable differences, as shown by [93–96].

### 3.2. Black, Elemental and Organic Carbon (BC, EC and OC)

EC specifically refers to materials constituted by carbon atoms only. Amorphous and graphitic EC were observed in particles emitted by combustion processes [80]. However, nearly all available EC data result from thermal and thermal-optical measurements (applied to PM samples collected on filters). Such measurements rely on EC's refractory property, i.e., EC does not volatilise or combust below given temperature thresholds. Various purely thermal and thermal-optical methods were used in recent decades. These methods generally lead to significantly different EC values. In all thermal and thermal-optical methods, EC is operationally defined. A European standard [97,98] for determining EC in $PM_{2.5}$ deposited on filters was recently developed (EN 16909), which is not applied worldwide.

BC has been used for years as a catch-all term to describe a variety of types of carbonaceous particles [99]. Nowadays, the definition of BC as the carbonaceous component of atmospheric particulate matter that absorbs all wavelengths of solar radiation present in the troposphere (280–2500 nm) is generally accepted [100]. BC mass concentrations cannot be directly measured. In contrast, a vast range of methods is available for determining the aerosol light absorption cross section, representing the aerosol "blackness". Absorption coefficient data ($M\ m^{-1}$) can be converted to equivalent black carbon (eBC) mass concentration data ($\mu g\ m^{-3}$) by applying an a priori chosen mass absorption cross section (MAC, $m^2\ g^{-1}$). Converting aerosol light absorption data to eBC means neglecting that not all light-absorbing aerosols are BC, and MAC values are highly variable with time and space [101]. In addition to "light absorption" methods, combined measurements of single-particle laser-induced incandescence can lead to refractory black carbon (rBC) mass concentrations by simultaneously determining the refractory mass of the incandescent particle and its chemical composition [102].

OC represents the bunch of C atoms belonging to organic molecules. In contrast with EC, organics can be volatilised in inert atmospheres at temperatures ranging from ambient to a few hundreds of Kelvin. Most OC compounds absorb IR and UV radiation strongly but are relatively transparent to visible (VIS, 400−700 nm) and near-IR (700−2500 nm) wavelengths. However, certain types of OC absorb radiation efficiently in the near-UV (300−400 nm) and VIS ranges [72].

### 3.3. Brown Carbon (BrC)

PM carbonaceous constituents can be classified according to their volatility and their light absorption spectra (Figure 6). These two properties are exploited by the main analytical methods implemented to address carbonaceous aerosols, namely thermal and optical methods. The class of carbonaceous PM which is not as refractory as EC and does not as ideally absorb all wavelengths (280 < λ < 2500 nm) as BC was named brown carbon (BrC) [103]. BrC consists of organic macromolecules with poorly characterised molecular structures, including fulvic acids, humic acids, and humic-like substances (HULIS) [104]. BrC constituents are characterised by their absorptivity, which increases more than proportionally with $\lambda^{-1}$, i.e., they proportionally absorb more short wavelengths as compared to BC.

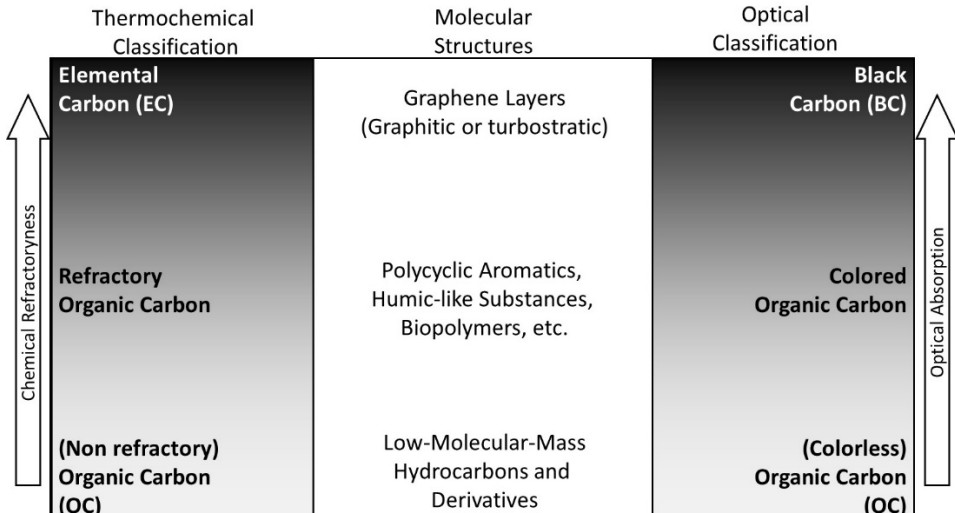

**Figure 6.** Optical and thermochemical classification of atmospheric carbonaceous particulate matter. BrC is an ensemble of light-absorbing (coloured) organic and relatively refractory macromolecules with a variety of molecular structures (diagram adapted from [72,105]).

The combustion of gasoline and diesel fuels under controlled laboratory conditions leads to BrC/BC ratios of approximately 1.7 [61]. A recent study showed that this ratio increases when the combustion temperature decreases, in association with an increase in non-refractory PAH mass fraction [62].

There is limited understanding of how particle optical properties—especially the contributions of BC and BrC—evolve with photochemical aging of smoke. Cappa et al. [106] analysed the evolution of the optical properties and chemical composition of particles produced from combustion of a wide variety of biomass fuels, largely from the western United States. The smoke is photochemically aged in a reaction chamber over atmospheric-equivalent timescales ranging from 0.25 to 8 days. In the USA, in a rural environment, Washenfelder et al. [107] found that the majority of BrC aerosol mass was associated with biomass burning, with smaller contributions from biogenically derived SOA. A recent study estimates primary BrC emission from biomass burning and biofuel use is globally 3.9 and 3.0 Tg year$^{-1}$, respectively, whereas secondary BrC is estimated to be ~5.7 Tg year$^{-1}$ [108,109]. Combined with BC emission, total BrC emission in China is estimated to be 3.42 Tg year$^{-1}$ in 2018 [110], of which 71% is from residential combustion, 14% is from vehicle exhaust, and 15% is from open biomass burning. Residential combustion is the main source of surface BrC in China, accounting for 60% on average, followed by open biomass burning (23%) and vehicle exhaust emissions (17%). Positive matrix factorisation (PMF) analysis showed that biomass burning, fossil fuel combustion, secondary formation, and fugitive dust are the major sources of BrC in the city, accounting for 55%, 19%, 16%, and 10% of the total BrC of PM$_{2.5}$, respectively [111]. In a previous work in smog chamber aging experiments of airborne biomass-burning particles [112], the authors characterised changes in the imaginary refractive index $k$ as a function of fuel type and aerosol aging state. They found that k and Absorption Ångström Exponent (AAE) were strongly correlated with the BC-to-OA ratio for a broad range of experiments using a variety of biomass fuels, indicating that the aerosol absorption coefficient depends more on the burn conditions than on fuel type. Using thermodenuder-based measurements, where semi-volatile organic compounds (SVOCs) were stripped off aerosols, they were able to demonstrate that BrC compounds in biomass-burning aerosol can be classified as extremely low-volatility organic compounds (ELVOCs, e.g., Murphy et al. [113]). They estimated that ELVOCs had an order of magnitude higher $k$ than the rest of the organic compounds.

There is a key challenge on BrC. As a result of the uncertainty surrounding the sources and properties of BrC, and its tendency to be co-emitted and mixed with BC, these aerosols are not well represented in atmospheric chemistry and climate models. Inclusion of BrC in

the population of OC aerosols changes the net effect of OC from scattering to close to zero, especially in areas with heavy biomass burning [114]. In a recent study from China, [115] corroborates the dominant role of BrC in total biomass burning absorption. Therefore, BrC is not optional but indispensable when considering the climate energy budget, particularly for biomass burning emissions (contained and open).

The measured light absorption of methanol-extractable OC derived from biomass burning (BB) and gasoline vehicle emissions exhibited strong wavelength dependence with AAE (light absorption coefficient at a given wavelength) values much higher than 2 [116]. Gasoline vehicles tend to emit stronger light-absorbing OC in winter than in summer. Compared to BB, the light absorption of OC from gasoline vehicle emissions was of the same magnitude but weaker, suggesting the importance of gasoline vehicle emissions as a BrC source in urban regions. Non-extractable OC accounted for a substantial part (~25%) of the total OC from gasoline vehicle emissions, and further study to measure its potential light-absorbing properties is warranted. Definitively, treating OA as non-absorbing particles would underestimate the radiative effect of OAs, especially in urban areas where motor vehicle emissions are a substantial fraction of the aerosol.

However, accounting for BrC in climate models can lead to counterintuitive effects [117]. Indeed, BrC aerosol contributes to positive forcing (warming) over bright terrain throughout the atmospheric aging timescales. However, with increased atmospheric residence time from 0 to 4.5 PED (PAM Equivalent Day, with PAM as Potential Aerosol Mass reactor), the integrated direct radiative forcing (DRF) efficiency decreases by approximately 27%, from $40.4 \pm 1.7$ to $29.4 \pm 2.8$ W m$^{-2}$. The PED is the time in days to reach the maximum aerosol mass produced in the PAM reactor. A corresponding decrease in DRF efficiency over ground is ~5%, from $-4.0 \pm 0.0$ to $-4.3 \pm 0.1$ W m$^{-2}$ for particle aging from fresh to 4.5 PED. Although approximately half of the solar spectrum's energy is distributed between 400 and 700 nm, $375-532$ nm forcing represents a significant warming potential over arctic terrain, providing additional momentum for climate imbalance. However, the change in optical properties at longer aging timescales implies that model-based estimates of warming due to BrC light absorption could be overestimated. Other recent results [118] showed that OC content can be an important contributor to light absorption when present in significant quantities (>0.9 OC/total carbon), source emissions have variable absorption spectra, and non-biomass combustion sources can be significant contributors to BrC. In a recent study on biomass burning, findings suggest that current modelled biomass burning particles contribute less to warming than previously thought, largely due to treatments of aerosol mixing state [55]. In addition, other findings illustrate that common atmospheric processes, such as evaporation and photochemical aging of BrC aerosol, can cause a red shift in absorbance, leading to a greater overlap with the solar spectrum [119].

Very few studies have investigated the BrC emissions from vehicles [120,121]. In a recent study [122], the concentrations, optical properties, and emission factors of OC, water-soluble OC (WSOC), and humic-like substances (HULIS) in fine particulate matter (PM$_{2.5}$) emitted from vehicles were studied in three road tunnels (the Wucun, Xianyue, and Wenxing tunnels in Xiamen, China). The mass concentrations and light absorption of OC, WSOC, and HULIS were higher at the exits of each tunnel than at entrances, demonstrating that vehicle emissions were a BrC source. At each tunnel's exit, the average light absorption contributed by HULIS-BrC to water-soluble BrC (WS-BrC) and total BrC at 365 nm was higher than the corresponding carbon mass concentration contributed by HULIS (HULIS-C) to WSOC and OC, indicating that the chromophores of HULIS emitted from vehicles had a disproportionately high effect on the light absorption characteristics of BrC. The emission factors (EFs) of HULIS-C and WSOC mass concentrations were highest at the Xianyue tunnel; however, the EFs of HULIS-BrC and WS-BrC light absorption were highest at the Wenxing tunnel, indicating that the chromophore composition of BrC was different among the tunnels and that the mass concentration EFs did not correspond directly to the light

absorption EFs. PAHs from vehicular emissions are also noted as a precursor of BrC in another research work [123].

### 3.4. Polycyclic Aromatic Hydrocarbons

Polycyclic aromatic hydrocarbons (PAHs) are characterised by organic compounds with multiple aromatic rings with possibly functional groups attached to these rings. PAHs co-emitted with BC during incomplete combustion are believed to be trapped in micro porous structure of BC, due to the high affinity of PAHs for flat aromatic surface [124]. PAHs are colourless, white, or pale to yellow organic compounds. There are several routes through which PAHs enter the environment and they are often found as a mixture. PAHs are a group of several hundreds of chemically related compounds with different structures and toxicity. These compounds are comprised of numerous individual compounds having at least two condensed rings. PAHs are divided into two categories based upon their molecular weight—those with less than four rings are termed as low-molecular-weight compounds, while those with four or more rings are high-molecular-weight compounds. According Mallah et al. [125], the effect of PAHs on human health is primarily based on the duration and route of exposure and the relative toxicity of PAHs. Elevated systolic blood pressure and pulse pressure are substantially correlated with PAHs.

Understanding the association between BC and PAHs is critical in order to determine their negative effects, transport and fate throughout the environment. BC is considered to be ubiquitous in the atmosphere and have a strong sorbtive power for organic pollutants such as POPs and PAHs. It has been revealed that the distribution of PAHs in different environmental sections is dependent upon BC distribution due to the strong sorption ability of BC for PAHs [126].

PAHs are fully involved in the formation of soot particles (Figure 7) although the early stages of soot formation, namely inception and growth, are highly debated and central to many ongoing studies in combustion research [127–129]. The combustion process of fuel starts with the decomposition reactions of the fuel molecules, i.e., pyrolysis and oxidation, and subsequent recombination and cyclisation reactions lead to the formation of PAHs. Recent results [130] contribute to the understanding of soot formation.

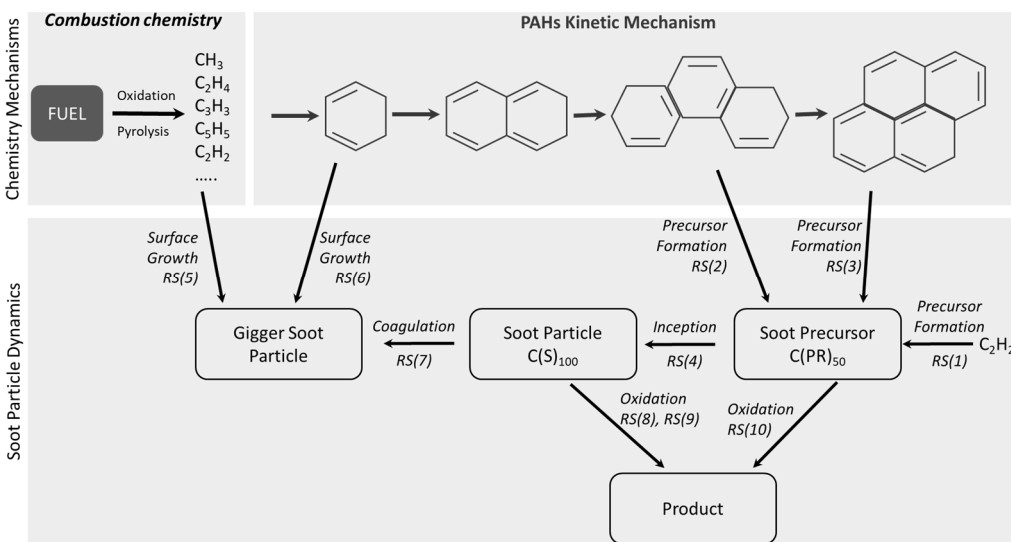

**Figure 7.** Soot formation pathways from fuel combustion adapted from several sources in the literature [127–129].

Four PAHs are particularly targeted in the legislation: benzo(a)pyrene, benzo(b)fluoranthene, benzo(k)fluoranthene and indeno(1,2,3-cd)pyrene (Aarhus Protocol on POPS). Benzo(a)pyrene and naphthalene contributed the most to the PAH carcinogenic potency of biomass-burning emissions according to [131].

*3.5. "Condensables" as a Contributor of Carbonaceous PM*

The US Environmental Protection Agency (EPA) [132,133] defines the concept of "condensable" PM (CPM) as:

*"Material that is vapor phase at stack conditions, but condenses and/or reacts upon cooling and dilution in the ambient air to form solid or liquid PM immediately after discharge from the stack. Note that all condensable PM is assumed to be in the $PM_{2.5}$ size fraction."*

By this definition, CPM is gaseous at the pre-discharge flue temperature, but it enters the particulate state after release to the atmosphere after some seconds or minutes. However, it can be noted that such definitions are not fixed, and especially for organic compounds CPM also include compounds that have indeed condensed in a cooling stage, but then quickly evaporate upon dilution [134]. CPM test methods can be divided into two categories: impinger cooling method and dilution cooling method. The concept of "condensables" is of major importance to understand the role of carbonaceous species from the emissions (high temperature and low dilution) to the ambient conditions (cooler temperature and high dilution). With respect to European emissions and the CLRTAP/EMEP, the issues have been addressed in a series of studies ([52,135] and references therein).

Historically, primary organic aerosol (POA) emissions have been assumed to be non-volatile and unreactive in atmospheric aerosol models. It has now been shown that in many cases, such assumptions are not correct [136]. Measurements of vehicular OA concentrations displayed significant negative (particle evaporation after collection on the filter) and/or positive artefacts/biases (vapour adsorption on the filter), providing strong hints about the semi-volatile nature of OAs [137,138]. For more than twenty years, dilution samplers have been used in research-grade studies to measure POA emission factors [139]. The use of these samplers was motivated by the semi-volatile character of primary emissions. Because of the complexity gap between research-grade datasets and promulgated test methods, emission inventories typically provide POA emissions using outdated assumptions and these data are post-processed at the aggregated sector level before being input to chemical transport models. Full consideration of the complexity of POA emissions in the design of test methods and emission inventories is needed to reduce uncertainty in the prediction of combustion source impacts.

Depending on the aim of the measurement, different sampling strategies are commonly used for product testing and health studies. They are presented in [140] clearly showing the impact of the method used to estimate the PM EF and how condensables can affect the measurement. The effects of physical phenomena were illustrated by [141], who developed a portable dilution system to measure aerosol emissions from stationary and mobile combustion sources. A considerable decrease of 37% (221–139 mg km$^{-1}$) was observed for light-duty diesel vehicle, with an increase in dilution ratio from 39:1 to 144:1. A similar decrease in particulate OC emission rates was observed, indicating scarcity of sorptive organics, and insufficient residence time for condensation limited the particle formation from vapour-phase organic compounds at high dilution ratios.

POA can be viewed in the broad sense as a continuum of carbon-containing species from gas-phase light VOCs to solid soot. Following the pioneering work of Donahue et al. and Robinson et al. [142,143] and subsequent studies, we can consider six categories sorted by decreasing volatility:

i.      Methane (the lightest VOC);
ii.     Non-methane VOCs (NMVOCs): usually molecules up to 12 carbons but higher HC could also be included;
iii.    Organic compounds of intermediate volatility (IVOCs);
iv.     Semi-volatile organic compounds (SVOCs) which can be split into SVOCs and LVOCs (low-volatility organic compounds);
v.      Extremely low-volatile organic compounds (ELVOCs), which are essentially non-volatile under relevant conditions;

 vi.  EC or BC—contains some atoms of hydrogen but is mainly composed of carbon. This species is usually not defined as an organic species.

 Categories (i) and (ii) can be always considered in the gas phase regardless of the dilution ratio and temperature. The definition of a volatile organic compound (VOC) varies from country to country. A general scientific definition of a VOC is an organic compound that evaporates or vaporizes under ambient conditions. These gases are emitted from various materials. From a legal point of view, the definition of VOCs can be different; for example, in the National Emission Ceiling directive: *volatile organic compounds' and 'VOC' mean all organic compounds arising from human activities, other than methane, which are capable of producing photochemical oxidants by reactions with nitrogen oxides in the presence of sunlight.*

 Categories (iii) and (iv) partition between the gas and particle phases depending on the temperature and dilution ratios. The notion of IVOCs is questionable and is addressed further in this review. These compounds, which have saturation concentrations between $10^3$ and $10^6$ µg m$^{-3}$, are termed intermediatevolatility organic compounds (IVOCs) by Donahue et al. [136]. The IVOCs category could overlap with the legal definition of VOCs since the degree to which IVOCs have been included with VOC sampling techniques historically is not clear. Recently, [144–146] has analysed and determined emission factors for IVOCs for different type of domestic fuels including wood.

 Categories (v) and (vi) can be considered as belonging exclusively to the particle phase. This phase is composed of heavy hydrocarbons and organic species. Species in categories (ii) to (iv) can react in the atmosphere and produce aerosol particles. The products of these reactions fall in categories (iv) to (v) by increasing their molar mass mainly due to the addition of oxygen atoms. A split in volatility classes has been proposed [113,147] based on saturation concentrations (Table 4). Therefore, organic condensables are species belonging to primary ELVOCs to SVOCs or secondary OA (SOA) categories.

**Table 4.** Volatility classes of VOCs [113,147].

| Description | Abbrev. | Saturation Concentration Range (µg m$^{-3}$) at 298 K | State in the Atmosphere |
|---|---|---|---|
| Extremely-low volatility | ELVOCs | $<3.2 \times 10^{-4}$ | Particle |
| Low volatility | LVOCs | $3.2 \times 10^{-4}$–$3.2 \times 10^{-1}$ | Mainly particle |
| Semi-volatile | SVOCs | $3.2 \times 10^{-1}$–$3.2 \times 10^2$ | PM and/or vapour phase |
| Intermediate volatility | IVOCs | $3.2 \times 10^2$–$3.2 \times 10^6$ | Vapour phase, readily oxidised to SVOCs |
| Volatile [†] | - | $>3.2 \times 10^6$ | Vapour phase |

[†] VOCs and NMVOCs are not listed here since they are superset, e.g., typically IVOCs and some SVOCs.

 We often speak about semi-volatile organic compounds (SVOCs) to name the organic species that would exist in both the gas phase and the particulate phase. In atmospheric chemistry, there can be different definitions of SVOCs. One study proposed to measure SVOCs based on mass collected on a "bare-Q" Quartz filter including the primary particle and some fraction of the semi-volatile organic vapour [148]. This definition was taken up by further works [137,138,149,150] to study the gas/particle partitioning of primary OAs. Compounds with an effective saturation concentration C* below $10^{-3}$–$10^{-4}$ µg m$^{-3}$ are generally captured with such a method. Robinson et al. [151] define SVOCs as presenting in particulate form at a low dilution of emissions corresponding to the conditions under which PM$_{2.5}$ emissions are measured in the US emission inventory. This definition has the disadvantage that the definition of SVOCs depends on the emission source, but the authors come to consider that compounds with a volatility less than or equal to $10^3$ µg m$^{-3}$ are SVOCs. Several other C*-based definitions of SVOCs exist in the literature, e.g., C* < 320 µg m$^{-3}$ [113].

 Robinson et al. [151] also introduced the concept of organic compounds with intermediate volatility (IVOCs) (detailed in Section 3.6). Indeed, the authors noticed that there was a "gap" between the VOCs identified by gas chromatography–mass spectrometry (GC–MS)—the speciation data for VOC emissions being largely based on this methodology)—with

$C^* \geq 107 \; \mu g \; m^{-3}$ and the SVOCs. The authors therefore designated as IVOCs the part of VOCs which could not be identified by GC–MS due to their low volatility. The authors postulated that IVOCs may be important precursors of SOA formation. In some modelling studies [135,143,152,153], IVOCs may have been presented as emissions missing from the inventory while some later studies (e.g., Jathar et al. [154] for vehicle emissions, though not for other sources) assume that VOC emissions were included in the NMVOC inventory, being that fraction which cannot be speciated.

### 3.6. S/IVOC Evolution and SOA Formation

SOA is formed in the atmosphere when oxidation of VOC (in a broad sense) species results in compounds of lower volatility that can partition to the particle phase. Gas-phase hydrocarbon autoxidation could be a rapid pathway for the production of in situ aerosol precursor compounds. Emerging evidence suggests that gas-phase hydrocarbon autoxidation is a highway to molecular growth and lowering of vapor pressure. This pathway produces hydrogen-bonding functional groups that allow a molecule to bind into a substrate. As stated by Rissanen [155], it is the crucial process in the formation and growth of atmospheric SOA.

The evolution of OA in the atmosphere can be depicted as in Figure 8. In the first seconds, after a plume leaves the stack or exhaust, freshly emitted primary organic species tend to condense due to rapid cooling, while subsequent dilution results in the evaporation of these POA. As the plume dilutes, reactions with oxidants on a large range of time-scales, and often over several chemical generations (e.g., VOCs to IVOCs to SVOCs), produce organic compounds with low volatility which can form SOA.

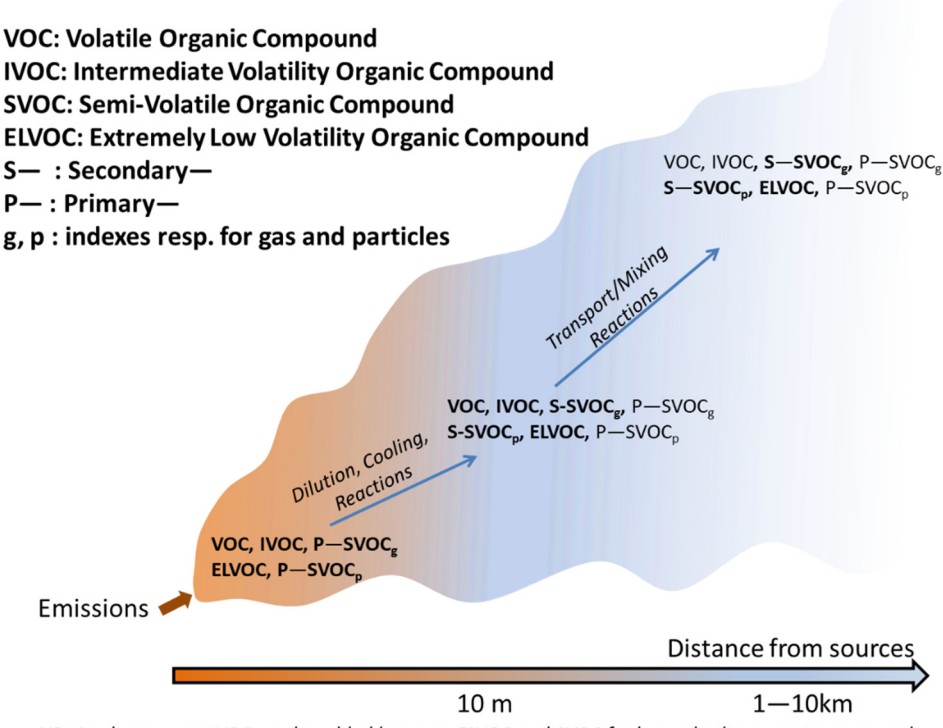

**Figure 8.** Schematic description of emission ageing in the atmosphere from a source.

The IVOC terminology has been introduced to describe compounds that were often not captured by the VOC measurement yet too volatile to be captured by a particle filter with a PUF sampling [151]. However, IVOCs can be measured by denuder [156]; these IVOCs can be viewed as gas-phase SVOCs that were not captured by the particle filter. Nonetheless, flame ionisation detection (FID) measurements, a common measure of total organic gases

for vehicular emissions, should contain all IVOCs [150,157,158]. However, as FID is often performed at a temperature approximately 190 °C all gases that evaporate above will be not taken into account. Additionally, flame ionisation detectors cannot detect inorganic substances and some highly oxygenated or functionalised species (aldehydes and ketones) such as infrared and laser technology can. Recently, barrier discharge ionisation detector (BID) for gas chromatography (GC–BID) was investigated by analysing different classes of organic compounds such as alcohols, alkanes, cycloaliphatic compounds, and polycyclic aromatic hydrocarbons (PAHs) [159]. The results obtained by GC–BID were compared with those of gas chromatography with flame ionisation detection (GC–FID), aiming to demonstrate the particular merits of the new BID detector over the well-established FID.

There is therefore no clear operational distinction between SVOCs and IVOCs. Moreover, the amount of SVOCs and IVOCs captured by the particle filter is a function of some experimental conditions (temperature and OA loading). Figure 9 illustrates the dependence of the particle-phase SVOCs on gas + particle SVOC ratio (or SVOC/(SVOC + IVOC) ratio) on the organic mass loading. Based on this figure, emissions of particles are dependent on experimental conditions and a part of SVOC emissions could be missing from emission inventories. If applicable dilution curves are known, they can be used by modellers to estimate SVOC/POA ratio for an emission factor by using temperature and organic mass loading.

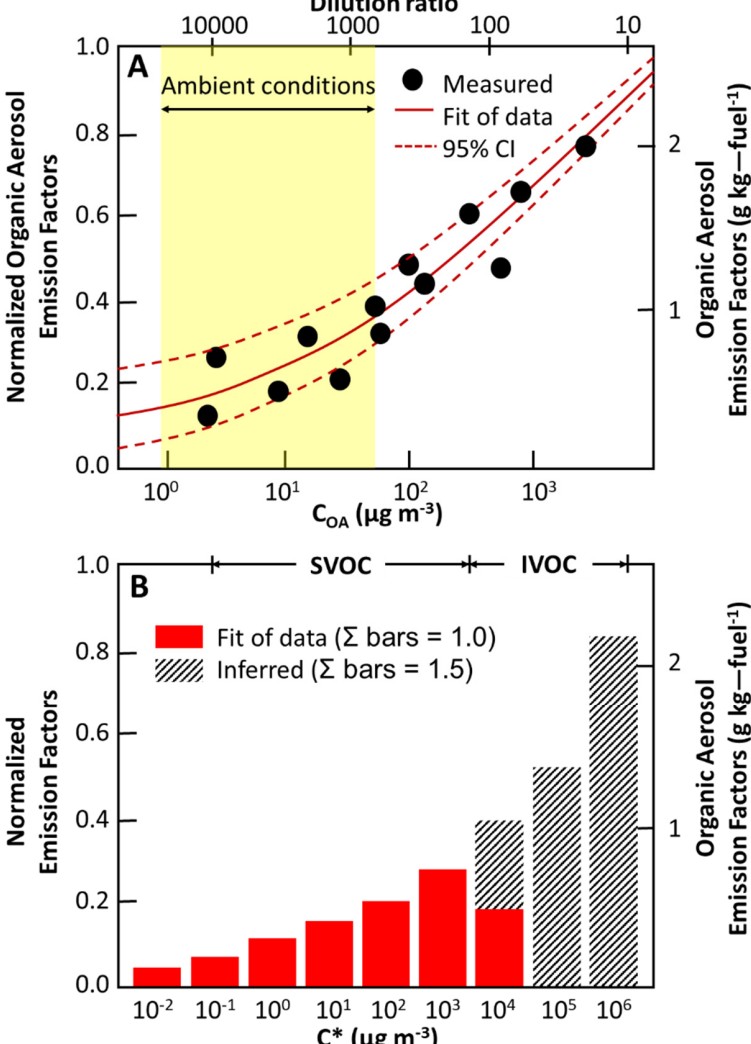

**Figure 9.** Normalised OA emission factor as a function of OA loading for a diesel motor exhaust at 300 K, adapted from Robinson et al. [151].

Based on the assumption that IVOCs are missing from inventory emissions, [151] introduced an IVOC/POA ratio to take into account those missing emissions. Actually, Robinson et al. [151] do not exactly say that there are missing emissions but just that IVOC emissions were not diagnosed in the inventory because they are part of the unresolved complex mixture (UCM) in speciation studies. In their paper, they report a ratio of 2.8 for emissions from a catalyst-equipped, gasoline-powered car and 1.5 for medium-duty diesel engine emissions based, respectively, on [156,160] and chose to use the 1.5 value. This ratio of 1.5 is used to account for IVOC species by most modelling teams working in this field [135,161–168] while other researchers [169] proposed to scale IVOC emissions on VOCs based on field campaigns conducted in an urban station in London.

The results reported in Schauer's publications [156,160] are representative of late 1990s American diesel vehicles and are therefore difficult to extrapolate to a recent fleet of European passenger cars. However, using measurements on French passenger vehicles [170] estimated that this factor of 1.5 may be well suited for passenger cars in France. However, this factor of 1.5 is certainly inappropriate as a presentation of the whole current vehicle fleet. It is interesting to note that the EMEP/EEA air pollutant emission inventory guidebook [47] only required reported emission factor to be measured on a filter at less than 52 °C. Reported emission factors could be sampled at cold, ambient and hot temperatures and therefore could be comprise a variable amount of SVOCs.

In China, recent results [171] showed that under low- and high-NOx conditions at different OA loadings, IVOCs contributed more than 80% of the predicted SOA. They built up a parameterisation method to estimate the vehicular SOA based on a bottom-up measurement of VOCs and IVOCs, which would provide more information when considering the vehicular contribution to the ambient OA. The results indicate that vehicular IVOCs contribute significantly to SOA, implying the importance of reducing IVOCs when making air pollution control policies in urban areas of China and the EU as well. The role of I/SVOCs has been highlighted in other studies, and the missing source may result from the oxidation of semi- and intermediate-volatility organic compounds and/or from anthropogenic molecules that undergo autoxidation or multiple generations of OH-initiated oxidation involving organic nitrates [172].

With the implementation of particle filters on diesel passenger cars, the amount of OAs emitted has decreased strongly and therefore, based on Figure 9, the fraction of SVOCs in the gas phase is expected to strongly increase due to the decrease in the OA loading in the dilution sampler. Any climate or regional historical model runs probably need to adjust for this over time.

The chemical and physical processes are commonly studied in smog chambers to calculate SOA formation yields and the effect of after-treatment systems (Figure 10). A recent study under dark conditions [173] showed the physical evolution of particles characterised during 6 to 10 h from Euro 3 to Euro 6 vehicles. An increase in particle mass was observed even without photochemical reactions due to the presence of intermediate-volatility organic compounds and SVOCs. These compounds were quantified at emission and induce a particle mass increase up to 17% $h^{-1}$, mainly for the older vehicles (Euro 3 and Euro 4). Condensation is 4-fold faster when the available particle surface is multiplied by 6.5. If initial particle number concentration is below $[8,9] \times 10^4 \, cm^{-3}$, a nucleation mode seems to be present but not measured by a scanning mobility particle sizer (SMPS). The growth of nucleation-mode particles results in an increase in measured [PN]. Above this threshold, particle number concentration decreases due to coagulation, up to $-27\%$ $h^{-1}$.

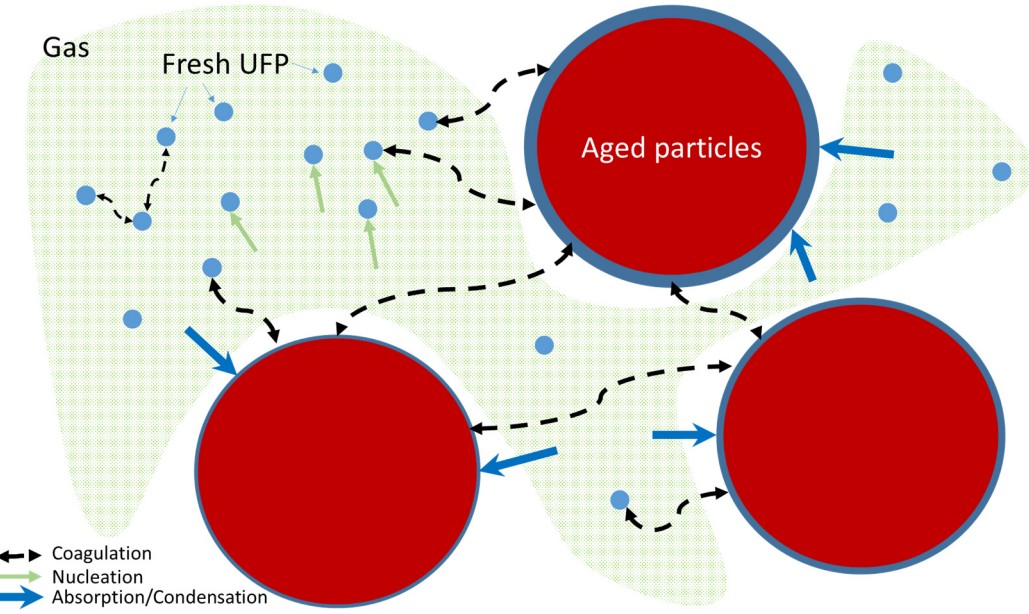

**Figure 10.** Physical processes at work during the dilution in the presence of pre-existing aged particles.

BrC could be a part of the low-volatility compounds which could be directly emitted or produced during the dilution process or formed later in ambient conditions (Figure 11) as discussed in [174] for nitrogen SOA. The "grey zone" corresponds to a list of compounds which could not be captured by a sampling technique or assigned to volatile or non-volatile species. The evolution of relative ratios between the mass concentrations of non-refractory organic species and BC is interesting—it varied between fuel types and displayed an inverse correlation with the modified combustion efficiency (MCE) of the burns [175], and a positive correlation with aging.

*Increasing volatility and/or lower C numbers*

*The grey zone?*

EC     BrC            SVOC/IVOC         NMVOC

OC/OM

BC1

BC2

Condensation/
Evaporation

SOA formation

*BC1 by thermo-optical or pure thermal methods*
*BC2 by pure optical methods*

**Figure 11.** Synoptic description of the carbonaceous species from graphitic carbon (EC) to NMVOCs.

One important finding [67] also illustrates that the role of the configuration of the dilution sampler was very important (i.e., open or closed transfer tube to the dilution tunnel;

both allowed in the current regulation) for a Euro 4 motorcycle. More representative actual particulate emissions of the motorcycle ($3 \times 10^{11}$ # km$^{-1}$) were found for the "open" method. The closed configuration resulted in lower solid particle emissions due to agglomeration (30%), but most importantly very high total particle emissions (i.e., including volatiles) (more than one order of magnitude) due to desorption of deposited material in the transfer tube to the dilution tunnel during high-speed driving. Thus, mixing directly at the tailpipe (e.g., open configuration or mixing tee) should be preferred for motorcycle measurements.

The issue of condensable organic aerosols is critical to calculate the emission factors of PM. Recently, a scientific project suggested EN-PME as a temporary test method for particles (similar to heat filter technique) from wood heating [176]. This method is now suggested as a test method for EN16510 and there is an ongoing verification project to validate this test method. The direct comparison of the EN-PME sampling method and NS3058 (based on dilution tunnel) in parallel from the current experimental campaign showed that emissions from the NS measurements are 11-fold higher with 108 mg MJ$^{-1}$ compared to the EN-PME test with 14 mg MJ$^{-1}$. The differences range between 2 and 60 times and are largely due to condensables.

In the literature, a compilation of emission factors [177] were derived from fresh smoke sampled at the source that had usually cooled to ambient temperature, but undergone low photochemical aging. Rather few studies have been focused on post-emission processing, and thus data for testing or constraining the chemical mechanism in smoke photochemistry models are very limited. Photochemical oxidation produced substantial new OA. Only a small fraction of this new OA can be explained using state-of-the-art SOA models and the measured decay of traditional SOA precursors. The application of models that explicitly track the partitioning and aging of low-volatility organics, including water-soluble species, should be compared to measurements with a suite of instruments in chambers. These kinds of models have been used as in previous studies [170], e.g., to relate elapsed times and measured concentrations of OC, and the condensation of SVOCs between the gas and particle phases is simulated with a dynamic aerosol model. Characteristic times to reach thermodynamic equilibrium between gas and particle phases may be as long as 8 min. Therefore, if the elapsed time is less than this characteristic time to reach equilibrium, gas-phase SVOCs are not at equilibrium with the particle phase and a larger fraction of emitted SVOCs will be in the gas phase than estimated by equilibrium theory, leading to an underestimation of emitted OC if only the particle phase is considered or if the gas-phase SVOCs are estimated by equilibrium theory.

At last, although it is indirectly linked with vehicle emissions, asphalt is also known to be a contributor to SOA formation through emissions of SVOCs and IVOCs [178]. Refuelling operations are also known to be an important source of VOCs/IVOCs leading to SOA formation, has and have been addressed in studies on Chinese cities [179]. These two issues are not within the scope of this review.

### 3.7. Ultrafine Particles

When inhaled, the scale of some of the nano-sised particles enables them to pass almost unheeded into the lungs and then even into the bloodstream [89]. The lack of efficient removal processes, coupled with the composition of the cocktail of adsorbed species, gives rise to environmental health concerns. Ultrafine particles (UFPs) are aerosols with an aerodynamic diameter of 0.1 μm (100 nm) or less. There is growing concern in the public health community about the contribution of UFPs to human health [180]. Despite their modest mass and size, they dominate in terms of the number of particles in ambient air. Large surface area and high surface reactivity enable UFPs to adsorb, for a given mass of PM, greater quantities of hazardous metals, and organic compounds that can generate oxidative stress. Harmful systemic health effects of PM$_{10}$ or PM$_{2.5}$ are often attributable to the UFP fraction. According to recent works [181,182], exposure to combustion-generated ultrafine particles and engineered nanoparticles can also be linked to neurodegenerative diseases documented in young, urbanised residents of polluted cities.

The Transport-Derived Ultrafines and the Brain Effects (TUBE) [183] project will increase knowledge on harmful ultrafine air pollutants, as well as semi-volatile compounds related to adverse health effects. By including all the major current combustion and emission control technologies, the TUBE project (funded by the European Union's Horizon 2020 research and innovation programme 2019–2023) aims to provide new information on the adverse health effects of current traffic, as well as information for decision makers to develop more effective emission legislation. Most importantly, the TUBE project will include adverse health effects beyond the respiratory system; TUBE will assess how air pollution affects the brain and how air pollution particles might be removed from the brain.

UFP are more and more recognised to have an impact on the climate though cloud formation and then precipitations regimes [184,185]. Thus, concurrent reductions in cloud droplet size modes by the introduction of excessive UFPs into the atmosphere result in diverse unwanted side effects, such as changes in the distribution and intensity of rainfall at a larger scale, causing either drought or flooding in extreme cases. Such drastic climate change affects the global hydrological cycle and thereby affects global public health both directly and indirectly.

At the exhaust pipe of a vehicle or in a plume of a heating system, UFPs can quickly evolve and/or disappear through deposition, condensation and coagulation processes, particularly for the smallest UFPs [186]. In clean environments, the nucleation of inorganic (sulphate, nitrate, ammonium) and organic compounds can be a secondary source of UFPs [90,187]. The size of 80–100 nm is the lower limit of the particle range at which particles have the highest time of residence usually between 0.1 μm and 1 μm. In this range of particles, the processes are known to be less efficient. However, very small UFP can have a very low time of residence due to coagulation processes with coarser particles [188]. However, a recent modelling study showed that at the street level, meteorological conditions were less impactful on particle number concentrations [189].

Patoulias and Pandis [190] found in a recent work that the addition of IVOC emissions and their aging reactions in a model led to a surprising reduction in the total number of particles, and $N_{10}$ (number of particles with a diameter less than 10 nm) by 10–15% and 5–10%, respectively, and to an increase in the concentration of $N_{100}$ (number of particles with a diameter less than 100 nm) by 5–10%. According to the authors, this occurred as a result of accelerated coagulation and reduced nucleation rates.

Giechaskiel et al. [191] recently published a review on particle number measurements for vehicle exhaust. The main finding of their overview is that, even though total particle sampling and quantification are feasible, details for their realisation in a regulatory context are lacking. It would be important to define the methodology details and conduct inter-laboratory exercises to determine the reproducibility of a proposed method.

The FASTER project (Fundamental Studies of the Sources, Properties and Environmental Behaviour of Exhaust Nanoparticles from Road Vehicles) is the most exhaustive recent work studying the size distribution—and, in unprecedented detail, the chemical composition—of nanoparticles sampled from diesel engine exhaust [192]. In this framework, Nikolova et al. [193] developed a model (CiTTy-Street-UFP) of traffic-related particle behaviour in a street canyon and in the nearby downwind urban background accounting for aerosol dynamics and the concentration of component organics. The modelled nucleation-mode peak diameter, which is 23 nm in the steady-state street canyon, decreases to 9 nm in a travel time of just 120 s. All modelled SVOCs in the sub-10 nm particle size range evaporate, leaving behind only non-volatile material. This evaporation effect was already reported in previous work for the same project with observations of a decrease in the median diameter by up to approximately 6 nm [194]. Deposition and coagulation on the aerosol distribution displayed a weak influence at this timescale, as reported by Harrison et al. [192]. In another study on aero-engines, the nucleation mode of the aerosol distribution at the exhaust dominates the total organic PM at idle, indicating that IVOCs/SVOCs may be significant in the homogeneous and heterogeneous nucleation process [195]. Additionally, Molteni et al. [196] explained that the fact that oxidation of aromatic HC can

rapidly form highly oxygenated molecules of very low-volatility makes aromatic HC a potential contributor to nucleation and early particle growth during nucleation episodes observed in urban areas in previous studies [197]. According to Liati et al. [76], particles smaller than ~10 nm were rarely detected in this transmission electron microscopy (TEM) study and are predominantly ash attached on or enclosed in soot.

## 4. Mitigation Measures for Road Traffic PM Emissions and Its Impact on Carbonaceous Emissions

In the following sections, a general overview of technologies in road transport and other key activity sectors to reduce emissions of particulate matter, focussing on BC and OM as well as PAHs and ultrafine particles (UFP), is proposed. This recent literature review highlights the role of technologies to abate emissions of particulate species.

### 4.1. Co-Benefits of Air Quality and Climate Change Mitigation Strategies

Recent contributions of the Task Force on Integrated Assessment Modelling (TFIAM) and the Task Force on Techno-Economic Issues (TFTEI) to the review of the Gothenburg Protocol in the frame of prioritising actions to reduce PM and BC emissions [22,198,199] underline that BC emission factors are still uncertain, and future research might come to change their results slightly. According to current legislation, there will be $PM_{2.5}$ and BC emission reductions, through absolute co-benefit technologies, of 73% and 79% between 2020 and 2030 in regions of the EU27 + Norway + Switzerland + United Kingdom. In the EECCA region, these figures are 36% and 64%, respectively; and in the non-EU-Balkan + Turkey region, 39% and 73%, respectively. In the EU, co-benefit technologies contribute to 2% of the emission reduction for both $PM_{2.5}$ and BC and are engine exhaust technologies in diesel engines and high-grade coal in stoves.

To improve communication and start developing a common understanding on BC, there is a clear need to develop simple metrics for BC, and establishing a "BC Footprint" concept, as proposed by a recent study [200], could be a first step. A BC Footprint would allow the comparison of different BC emission sources and levels of atmospheric BC concentrations, and would enable more efficient communication regarding the climate, health, and air quality impacts of BC. A recent analysis [201] confirms that for mitigation to be rapidly effective at reducing the rate of global warming, targeting BC emissions (where they are not co-emitted, e.g., with $SO_2$) would be efficient but with a low final payoff. However, in a recent study [202], it was shown that the decline in surface air temperatures with reduced BC emissions is weaker than would be expected from the magnitude of its instantaneous radiative forcing at the top of the atmosphere.

To explore the conceivable range of future air quality and, in particular, of population exposure to $PM_{2.5}$, which has been associated with the most harmful health impacts, a recent study [203] develops a series of alternative emission scenarios up to 2040 at the global level. As described in this study, in addition to improved human health (SDG 3—Improve human health and well-being), particularly in urban areas (SDG 11—Sustainable cities and communities), a clean-air scenario would deliver a host of co-benefits in other policy areas through several pathways. For mitigating climate change (SDG 13—climate action), some of the measures will not only reduce emissions of $PM_{2.5}$ precursors but will simultaneously reduce emissions that contribute to temperature increase. In particular, $CO_2$ emissions of the clean-air scenario will be approximately 40% lower than in the reference case in 2040, $CH_4$ 33%, and particularly BC by 90%.

Another recent study [204] contributed with an integrated assessment model-based scenario analysis of BC-focused mitigation strategies aimed at maximising air quality and climate benefits. The impacts of these policy strategies have been examined under different socioeconomic conditions, climate ambitions, and BC mitigation strategies. The study finds that measures targeting BC emissions (including reduction of co-emitted OC, sulphur dioxide, and nitrogen dioxides) result in a significant decline in premature mortality due to ambient air pollution, in the order of 4 to 12 million avoided deaths between 2015 and 2030.

Under certain circumstances, BC mitigation can also reduce climate change, i.e., mainly by lowering BC emissions in the residential sector and in high-BC-emission scenarios. Still, the effect of BC mitigation on global mean temperature is found to be modest at best (with a maximum short-term global mean temperature decrease of 0.02 °C in 2030) and could even lead to warming (with a maximum increase of 0.05 °C in case of a health-focused strategy, where all aerosols are strongly reduced). At the same time, strong climate policy would improve air quality through reduced fossil fuel use, leading to an estimated 2 to 5 million avoided deaths in the period up to 2030. By combining both air quality and climate goals, net health benefits can be maximised.

A recent estimate [205] showed the projected tons of BC emissions avoided under recently adopted policies and the potential to further reduce BC emissions by accelerating the global implementation of soot-free standards for vehicles, engines and fuels. The study also evaluates the implications for global temperature pathways and societal costs that include climate and health damages. The authors estimate that currently adopted policies will reduce global on-road diesel BC emissions to 40% below 2010 levels by 2030. They show projected global diesel BC emissions for five policy scenarios in comparison, with a 75% reduction in global diesel BC emissions from 2010 to 2030, corresponding to the level of BC reduction targeted by the CCAC Scientific Advisory Panel [206]. Policies that have been adopted or implemented since 2015 are projected to avoid 2 million tons of diesel BC emissions cumulatively from 2015 to 2030, equivalent to a 16% reduction in cumulative emissions compared with a baseline without these policies. More than 70% of these BC reductions are attributable to soot-free standards in China and India. Nevertheless, currently adopted policies are still insufficient to achieve a 75% reduction in global diesel BC emissions from 2010 to 2030.

Shindell et al. [207] showed that a major barrier to emission reductions is the difficulty of reconciling immediate, localised costs with global, long-term benefits. However, 2 °C trajectories not relying on negative emissions or 1.5 °C trajectories require the elimination of most fossil-fuel-related emissions. This generally reduces co-emissions such as vehicular emissions that cause ambient air pollution, resulting in near-term, localised health benefits. They examined the human health benefits of increasing 21st-century $CO_2$ reductions by 180 GtC, an amount that would shift a 'standard' 2 °C scenario to 1.5 °C or could achieve 2 °C without negative emissions. The decreased air pollution leads to 153 ± 43 million fewer premature deaths worldwide, with ~40% occurring during the next 40 years, and minimal climate disbenefits. More than one million premature deaths would be prevented in many metropolitan areas in Asia and Africa, and >200,000 in individual urban areas on every inhabited continent except Australia.

### 4.2. After-Treatment Systems

After-treatment systems of vehicle exhaust are described in many studies and reference papers [208]. Catalytic exhaust-gas after-treatment using a three-way catalytic converter (TWC) is currently the most effective form of emission control for gasoline engines. The TWC is an integral component of the exhaust-emission control systems of both manifold-injection engines and gasoline direct-injection engines. Although these ideal conditions cannot always be maintained in all operating states, pollutant emissions can on average be reduced by more than 98%. The TWC is not directly involved in the suppression of direct PM and soot emissions but can have an impact on secondary formation of PM.

The selective catalytic reduction (SCR) of NOx with aqueous urea ("urea SCR") is originally a steady-state technology used in stationary source NOx control [209]. Catalysts for this application are usually quite large, and typically consist of lower-cost materials such as vanadium/titanium or Fe/zeolite. The SCR catalyst must function and survive in a multi-component system that also controls hydrocarbons, carbon monoxide, and soot, and faces poisoning by sulphur, phosphorus, and a variety of other elements present in fuel, oil, and any upstream component including other catalysts.

The diesel oxidation catalyst (DOC) is a non-filter-based open monolith (flow-through) system resembling the conventional catalytic converters for gasoline and diesel engines with some significant variation of the catalyst composition so as to optimize the catalyst activity under lean conditions. The noble metals are impregnated into a highly porous alumina washcoat approximately 20–40 mm in thickness that is applied to the passageway walls. Most of the DOCs used in the international market contain platinum (Pt) and palladium (Pd).

The diesel particulate filter (DPF) is usually a cylindrical ceramic structure with thousands of small parallel channels positioned in the longitudinal direction of the exhaust system. The porous surface wall-flow monoliths are made of ceramics with higher and more precisely controlled porosity, and the adjacent channels in the wall-flow filters are alternatively plugged at each end, thus forcing the diesel aerosol to flow through the porous substrate walls, which act as the filter medium. Particles that are too coarse to pass through the porous surface are physically collected and stored in the channels. The DPF walls are built to have an optimum porosity, enabling the exhaust gas to pass through the walls without much hindrance, and to be sufficiently impervious regarding collecting the particulate species.

The gasoline particulate filter (GPF) technology is an adaptation of DPF to gasoline vehicles. With increasingly stringent emission regulations, research regarding the application of PM filters in gasoline direct injection (GDI) engines will become an inevitable trend. The particulate filter must have high filtration efficiency and a long life and satisfy other requirements, and the flow resistance (pressure drop) must be small [210]. The particulate filter is composed of nested foam metal cylinders and annular diversion channel plugs.

In a catalysed DPF (CDPF), a catalytic material, mostly with low-level platinum group metal, is coated onto the surface of the filter to lower the ignition temperature necessary for oxidising accumulated PM in the 300–400 °C range, allowing the filter to self-regenerate during the periods of high exhaust gas temperature. CDPFs achieve over 90% reduction in PM and soot as well as in HC and CO. CDPFs exhibit excellent PM filtration efficiencies and are characterised by inherent relatively high pressure drop. The feasibility of soot combustion depends to a great extent on the catalyst–soot contact conditions, and it is therefore necessary to maximize the interaction between the soot particles and the catalyst, both of which are solid materials.

A pioneering approach to maximize the soot–catalyst contact was implemented by a car maker in early 2000s, whose key component is a Ce-fuel additive that was reliably implemented in several million vehicles up to now. These vehicles have an active on-board additive system with its own tank-pump device that allows to dose the proper amount of Ce fuel additive to the diesel fuel. This metal organic compound in fuel leads to the formation of $CeO_2$ particles well embedded in the structure of diesel particulate, and thus in very good contact with the soot particles (intimate soot–catalyst contact). Therefore, lowers ignition temperatures can be reached by catalytic means, with the benefit of post-injected fuel savings [211]. A recent publication on the transformation of cerium oxide nanoparticles from a diesel fuel additive during the combustion in a diesel engine showed potential environmental and toxicological impacts [212] through a change of size and particle morphology.

All these devices can be combined. Typically, there are two different conceptual architectures for combining the SCR and DPF technologies to achieve a desired level of emission reduction performance: either SCR is placed upstream of DPF, i.e., DOC + SCR + DPF, or SCR is placed downstream of DPF, i.e., DOC + DPF + SCR.

*4.3. Carbonaceous Composition of PM Emissions at the Exhaust Pipe*

Exhaust PM mainly consists of EC, OC and inorganic components including metallic ash and ions. Therefore, different literature values have been collected and average EC and OC values have been proposed by [213]. The variability of the data collected from tunnel, roadway and dynamometer studies, and the uncertainties in the measurement of, in

particular, OC, indicate that exhaust PM speciation is bound to be highly uncertain. Because of this uncertainty, mean EC and BC values are considered practically equal [214,215]. Although it is known that EC and BC definitions and determination methods differ, this is considered to be of inferior importance compared to the overall uncertainty in determining either of them per vehicle emission control technology.

Despite overall uncertainties, reliable BC/OC ratios can be developed, because there is a general agreement in the measurements from tunnel and laboratory studies with regard to the emission characteristics of diesel and petrol vehicles. The effect of different technologies (e.g., oxidation catalyst and diesel particle filter) on emissions is also rather predictable.

Table 5 from Ntziachristos and Samarras [216] suggests ratios between organic material (OM) and BC (OM/BC) and BC/$PM_{2.5}$ (both expressed as percentages) that can be applied to the exhaust PM emissions for different vehicle technologies. 'Organic material' is the mass of OC corrected for the hydrogen and other atoms content of the compounds collected. The sources of these data, and the methodology followed to estimate these values, is given in [213]. An uncertainty range is also proposed, based upon the values in the literature. The uncertainty is in percentage units, and is given as a range for both ratios proposed. For example, if the OM/EC ratio for a particular technology is 50% and the uncertainty is 20%, this would mean that the OM/EC ratio is expected to range from 40% to 60%. This is the uncertainty expected on fleet-average emissions, and not on an individual vehicle basis. Individual vehicles in a specific category may exceed this uncertainty range. The ratios also correspond to average driving conditions, with no distinction between driving modes or hot and cold-start operation.

**Table 5.** $PM_{2.5}$ emission factors, split of PM in elemental (BC) and organic mass (OM) from Ntziachristos and Samarras [216].

| Category | Euro Standard | $PM_{2.5}$ EF (mg km$^{-1}$) | BC/$PM_{2.5}$ (%) | OM/BC (%) | Uncertainties (%) |
|---|---|---|---|---|---|
| Petrol PC and LCV | PRE-ECE | 2.2–2.3 | 2 | 4900 | 50 |
| | ECE 15 00/01 | 2.2–2.3 | 5 | 1900 | 50 |
| | ECE 15 02/03 | 2.2–2.3 | 5 | 1900 | 50 |
| | ECE 15 04 | 2.2–2.3 | 20 | 400 | 50 |
| | Open loop | 2.2–2.3 | 30 | 233 | 30 |
| | Euro 1 | 2.2–2.3 | 25 | 250 | 30 |
| | Euro 2 | 2.2–2.3 | 25 | 250 | 30 |
| | Euro 3 | 1.1–2.2 | 15 | 300 | 30 |
| | Euro 4 | 1.1 | 15 | 300 | 30 |
| Diesel PC and LCV | Conventional | 220.9–356 | 55 | 70 | 10 |
| | Euro 1 | 84.2–117 | 70 | 40 | 10 |
| | Euro 2 | 54.8–117 | 80 | 23 | 10 |
| | Euro 3 | 39.1–78.3 | 85 | 15 | 5 |
| | Euro 4 | 31.4–40.9 | 87 | 13 | 5 |
| | Euro 3,4,5 equipped with DPF and fuel additive | - | 10 | 500 | 50 |
| | Euro 3,4,5 equipped with a catalysed DPF | - | 20 | 200 | 50 |
| Diesel HDV | Conventional | 333–491 | 50 | 80 | 20 |
| | EURO I | 129–358 | 65 | 40 | 20 |
| | EURO II | 61–194 | 65 | 40 | 20 |
| | EURO III | 56.5–151 | 70 | 30 | 20 |
| | EURO IV | 10.6–26.8 | 75 | 25 | 20 |
| | EURO V | 10.6–26.8 | 75 | 25 | 20 |
| | EURO VI | 0.5–1.3 | 15 | 300 | 30 |

**Table 5.** *Cont.*

| Category | Euro Standard | PM$_{2.5}$ EF (mg km$^{-1}$) | BC/PM$_{2.5}$ (%) | OM/BC (%) | Uncertainties (%) |
|---|---|---|---|---|---|
| L-Categories ** | Conventional 2 stroke | 176 | 10 | 900 | 50 |
| | Euro 1 2 stroke | 45 | 20 | 400 | 50 |
| | Euro 2 2 stroke | 26 | 20 | 400 | 50 |
| | Conventional 4 stroke | 14–176 | 15 | 560 | 50 |
| | Euro 1 4 stroke | 14–40 | 25 | 300 | 50 |
| | Euro 2 4 stroke | 3.5–7 | 25 | 300 | 50 |
| | Euro 3 4 stroke | 0.96–4 | 25 | 250 | 50 |

Notes: The values originate from available data in the literature and engineering estimates of the effects of specific technologies (catalysts, DPFs, etc.) on emissions. The estimates are also based on the assumption that low-sulphur fuels (<50 ppm t. S) are used. Hence, the contribution of sulphate to PM emissions is generally low. In cases where advanced after-treatment is used (such as catalysed DPFs), then EC and OM does not add up to 100%. The remaining fraction is assumed to be ash, nitrates, sulphates, water and ammonium salts. ** L-categories are *Mopeds and Motorbikes*.

The OM/BC ratio is larger for gasoline vehicles and declines from old to modern vehicles. For diesel HDV EURO VI, it is noteworthy to see the drop of BC/PM$_{2.5}$ ratio with a surge of the OM/BC ratio certainly due to soot mostly removed by the DPF and the dominance of OM formed during the cooling. They are associated to the lowest PM$_{2.5}$ ELV.

*4.4. Non-Exhaust Emissions*

Direct emissions from brakes, tyres and road surface wear are the major sources of non-exhaust emission (NEE). While the exhaust emissions decrease continuously due to improved exhaust after-treatment and filter technologies, the NEE share is increasing because of rising vehicle mileage. According to the recent results of the European Environment Agency (EEA), a significant fraction of 33.9% and 26.7% of the road transport-related PM is assigned to automobile tyre and brake wear for PM$_{10}$ and PM$_{2.5}$, respectively. Further studies consider brake wear separately and report contributions of 21% to the traffic-related PM$_{10}$ in urban areas [217]. Brake wear was also reported to give 16%–55% by mass to total non-exhaust traffic-related PM$_{10}$ emissions in urban environments [218]. A complete review of NEE is provided in Harrisson et al. [219].

The dedicated EEA guidebook chapter [220] covers emissions of particulate matter (PM) including BC which are due to road vehicle tyre and brake wear (NFR code 1.A.3.b.vi), and road surface wear (NFR code 1.A.3.b.vii), as depicted in Table 6. The focus is on primary particles—in other words, those particles emitted directly as a result of the wear of surfaces—and not those resulting from the resuspension of previously deposited material. Resuspension of dust road depends of the site climatology, traffic flow and meteorological parameters. As part of resuspended dust can be anthropogenic and biogenic in origin and remain difficult to disentangle from direct emissions of brakes, tyre and road wear, we must be cautious to avoid double counting. Consensus exists in the scientific literature that non-exhaust emissions are becoming the dominant source of PM from road traffic and that PM [219], and particularly PM$_{2.5}$. Despite these demonstrated negative externalities, non-exhaust emissions have been only tangentially addressed by public policies to date. Given the magnitude of the aggregate social costs they entail, and the fact that the transition to electric vehicles will not lead to significant reductions in non-exhaust emissions, policy makers should invest resources in determining how to optimally reduce them via targeted policy instruments [221].

**Table 6.** Emission factors ranges for vehicles from two-wheeled light heavy-duty vehicles according to the Tier 2 methodology [220].

|  | **Tyres** | **Brakes** | **Road Wear** |
|---|---|---|---|
| *TSP EF (mg km$^{-1}$)* | 4.6–16.9 | 3.7–11.7 | 6–15 |
| *Fraction (%)* | | | |
| *TSP* | 100 | 100 | 100 |
| *PM$_{10}$* | 60 | 98 | 50 |
| *PM$_{2.5}$* | 42 | 39 | 27 |
| *PM$_1$* | 6 | 10 | - |
| *PM$_{0.1}$* | 4.8 | 8 | - |

According to Baensch-Baltruschat et al. [222] (and references therein), tyre wear contribution to PM$_{10}$ was reported to be up to approx. 11 mass %. The highest absolute concentration of tyre wear in ambient air is reported to be 3.4 μg m$^{-3}$ (without mass contribution from road wear). The BC fraction is on average equal to 15% of TSP for tyre and brake wear in other recent studies [223,224]. PM$_{10}$ usually displays a unimodal mass size distribution with maxima between 2 and 6 μm, with on average a PM$_{10}$ EF of 6.7 mg km$^{-1}$ vehicle$^{-1}$. Particle number distributions of brake wear PM$_{10}$ appear to be bimodal with both peaks lying within the fine mode. Most researchers report one peak of the distribution being among ultrafine particles (<100 nm), while others find it at somewhat bigger sizes (approximately 300 nm). The most important chemical constituents of brake wear are Fe, Cu, Ba and Pb. OC is also present in significantly higher concentrations compared to EC. On the other hand, there is very limited information regarding specific organic constituents of brake wear PM$_{10}$. In addition, zum Hagen et al. [225] found a particle number PN emission factor of approximately $4.9 \times 10^{10}$ km$^{-1}$ brake$^{-1}$ estimated for realistic vehicle brake temperatures.

A recent study investigated [226] the content of BC in brake wear particles. The results verified the existence of BC emission from disc brake system. Brake pad surface treatment and graphite content also have strong influence on BC emission of disc brake contact. A scorched brake material features lower BC and particulate matter emissions than non-scorched brake materials. Meanwhile, high graphite content in the brake material tends to expedite BC emission. BC emission shows a proportional correlation with PM$_1$ levels from disc brake system (approximately 20–30% of PM$_1$). The fraction of BC in PM$_1$ depends on the surface condition and graphite content of the brake materials.

Farwick zum Hagen et al. [217] studied on-road vehicle measurements of brake wear particle emissions. For the conventional brake pad material, the TSP EF was in the range of 1.4–2.1 mg km$^{-1}$ brake$^{-1}$, while the use of a novel material composition showed approximately 18% lower PM$_{10}$ emissions. Regarding particle numbers, the EF reached total numbers of $2 \times 10^{12}$–$1.3 \times 10^{13}$ km$^{-1}$ brake$^{-1}$, which was dominated by UFP emissions during high-brake-temperature sections and included volatile particles. The novel material produced approximately 60% less particles. However, the PN emissions were obtained during unrealistic high-temperature sections and were not representative for realistic driving. For temperatures observed at the reference brake, the PN EF would not exceed $1 \times 10^{10}$ km$^{-1}$ brake$^{-1}$. The critical brake temperature at which UF emission occurred was found at 168 and 178 °C for the conventional and novel material, respectively. The temperature of the reference brake did not exceed 153 °C during the same test, thus UFP brake emissions are not expected during normal driving.

Mathissen et al. [227] studied the potential generation of ultrafine particles from the tyre road interface was investigated during real driving. An instrumented sport utility vehicle equipped with summer tyres was used to measure particle concentrations with high temporal resolution inside the wheel housing while driving on a regular asphalt road. Different driving conditions, i.e., straight driving, acceleration, braking, and cornering, were applied. For normal driving conditions, no enhanced particle number concentration

in the size range 6–562 nm was found. Unusual manoeuvres associated with significant tyre slip resulted in measurable particle concentrations. The maximum of the size distribution was between 30 and 60 nm. An exponential increase in the particle concentration with velocity was measured directly at the disc brakes for full stop brakings.

A dilution tunnel was designed by Mamakos et al. [228] for the characterisation of brake-wear particle emissions up to 10 μm on a brake dyno. The particulate matter emission levels from a single front brake were found to be 4.5 mg km$^{-1}$ (1.5 mg km$^{-1}$ being smaller than 2.5 μm) over a novel real-world brake cycle, for a commercial Economic Commission for Europe (ECE) pad. Particle number (PN) emissions as defined in exhaust regulations were in the order of 1.5 to $6 \times 10^9$ # km$^{-1}$ brake$^{-1}$. Concentration levels could exceed the linearity range of full-flow condensation particle counters (CPCs) over specific braking events, but remained at background levels for 60% of the cycle. Similar concentrations were measured with condensation and optical counters, suggesting that most of emitted particles were larger than 300 nm. Application of higher braking pressures resulted in elevated PN emissions and the systematic formation of nano-sised particles that were thermally stable at 350 °C. Volatile particles were observed only during successive harsh braking events leading to elevated temperatures. The onset depended on the type of brakes and their history, but always at relatively high disc temperatures (280 to 490 °C).

A modelling study presented simulations of BC concentrations in a Parisian suburban street network using the SinG (Street in Grid) model [19]. The study investigated the effect of non-exhaust emissions on BC concentrations in streets, presenting a sensitivity analysis of wear emission factors and a new approach to estimate particle resuspension while respecting mass conservation at the street surface. The BC concentrations in streets proved to largely influence urban background concentrations. The two-way dynamic coupling leads to an increase in BC concentrations in large streets with high traffic emissions (with a maximal increase of approximately 48%) as well as a decrease in narrow streets with low traffic emissions and low BC concentrations (with a maximal decrease of approximately 50%).

To tackle brake emissions, a company has developed a brake particles collection system named TAMIC [229]. This device was designed to trap at least 80% of brake particles directly at the pad-disc interface without altering braking efficiency. It is composed of a brake calliper especially designed for the integration of grooved pads and of an aspiration system where brake particles are trapped. The aspiration system relies on pipes connected to a turbine equipped with a high-efficiency filter. Mass and number collection efficiencies have been assessed on brake rigs. A mass efficiency above 85% and a number efficiency up to 90% have been achieved. Performances on vehicle and impact on its behaviour have also been verified on test track and in real driving conditions with the same level of collection performances. The total number concentration of the PM was found by Mulani et al. [230] to be inversely linked to the disc's thermal conductivity, with the highest concentrations seen in friction pair with ceramic discs. The thermal conductivity of the disc is a crucial parameter in the brake pad design process.

To tackle tyre emissions, the Tyre Collective [231] aims to reduce these emissions by capturing the tyre particles when emitted. The device is fitted to the wheel and uses electrostatics to collect particles as they are emitted from the tyres, by taking advantage of various air flows around a spinning wheel. They claim that their prototype can collect 60% of all airborne particles from tyres, under a controlled environment on their test rig.

To complement our analysis, emissions from solvents in screen wash and de-icers now dominate VOC emissions from traffic in the UK, and exhibit a very different composition to exhaust VOC emissions [219]. VOC emission from brake emissions is very scarce but has been highlighted by recent works [232], showing a large contribution of phenols, aliphatic and aromatic hydrocarbons. Evaporative emission is also a source of VOCs, these species can produce SOA by chemical reactions.

*4.5. Impact of After-Treatment Systems on Particulate Matter Emissions*

In the last 30 years, diesel engines have made rapid progress to increased efficiency, environmental protection and comfort for both light- and heavy-duty applications [233]. The technical developments include all issues from fuel to combustion process to exhaust gas after-treatment. Diesel engine technologies representative of real-world on-road applications can be highlighted. Modifications of internal engine now make it possible to reduce particulate and nitrogen oxide emissions with nearly no reduction in power. Among these modifications are cooled exhaust gas recirculation, optimised injections systems, adapted charging systems and optimised combustion processes with high turbulence. With introduction and optimisation of exhaust gas after-treatment systems, such as the diesel oxidation catalyst and the diesel particulate trap, as well as NOx-reduction systems, pollutant emissions have been significantly decreased. Today, sulphur poisoning of diesel oxidation catalysts is no longer considered a problem due to the low-sulphur fuel used in Europe.

In the future, there will be an increased use of biofuels, which generally have a positive impact on the particulate emissions and do not increase the particle number emissions. Since the introduction of the EU emissions legislation, all emission limits have been reduced by over 90%. Further steps can be expected in the future. Retrospectively, the particulate emissions of modern diesel engines with respect to quality and quantity cannot be compared with those of older engines. Internal engine modifications lead to a clear reduction in particulate emissions without a negative impact on the particulate-size distribution towards smaller particles. The residual particles can be trapped in a diesel particulate trap independent of their size or the engine operating mode. The usage of a wall-flow diesel particulate filter leads to an extreme reduction of the emitted particulate mass and number, approaching 100%. A reduced particulate mass emission is always connected to a reduced particle number emission.

A recent review [234] covers the state-of-the-art of DPF technologies, including the advanced filter substrate materials, the novel catalyst formulations, the highly sophisticated regeneration control strategies, the DPF uncontrolled regenerations and their control methodologies, the DPF soot loading prediction, and the soot sensor for the PM on-board diagnostics (OBD) legislations. The progress of the highly optimised hybrid approaches, which involves the integration of diesel oxidation catalyst (DOC) + (DPF, NOx reduction catalyst), the selective catalytic reduction (SCR) catalyst coated on DPF, as well as DPF in the high-pressure exhaust gas recirculation (EGR) loop systems, is well discussed. The high-efficiency gasoline particulate filter (GPF) technology is being required to effectively reduce the PM and particulate number (PN) emissions from the gasoline direct-injection (GDI) engines to comply with the future increasingly stricter emissions regulations. The use of GPF confirmed its efficiency to reduce the PN in a recent work [235], a Euro 6d-Temp GDI vehicle with a GPF was tested on the road and in the laboratory with cycles simulating congested urban traffic, dynamic driving, and towing a trailer uphill at 85% of maximum payload. In this study, the ambient temperatures covered a range from −30 to 50 °C. The solid PN emissions were 10-fold lower than the PN limit under most conditions and temperatures. Only dynamic driving that regenerated the filter passively, and for the next cycle resulted in relatively high emissions although they were still below the limit. The results of this study confirmed the effectiveness of GPFs in controlling PN emissions under a wide range of conditions.

The temperature of diesel exhaust gas has an important effect on reducing pollutant emissions. Besides catalyst type, space velocity of exhaust gas, and emission form are the other parameters affecting the efficiency. With the after-treatment emission control systems, it is possible to reduce the damage of the pollutant emissions on air pollution, to meet emission standards and requirements, and to prevent the harmful effects of pollutant emissions on environment and human health [236].

A recent study [237] was performed focussing on (i) ultrafine particles, BC, BTEX, PAHs, carbonyl compounds, and $NO_2$ emissions from Euro 4 and Euro 5 diesel and gasoline passenger cars, (ii) the effect of driving conditions (e.g., cold start, urban, rural and

motorway conditions), and (iii) the impact of additive and catalysed DPF devices on vehicle emissions. The results showed that compared to hot-start cycles, cold-start urban cycles increased all pollutant emissions by 2 fold. The sole exception was $NO_2$, which was reduced 1.3–6 fold. Particulate and BC emissions from the gasoline engines were significantly higher than those from diesel engines equipped with DPF. Moreover, the catalysed DPF emitted approximately 3–10-fold more carbonyl compounds and particles than additive DPF, respectively, during urban driving cycles, while the additive DPF vehicles emitted 2- and 5-fold more BTEX and carbonyl compounds during motorway driving cycles. Regarding particle number distribution, the motorway driving cycle induced the emission of particles smaller in diameter (mode at 15 nm) than the urban cold-start cycle (mode at 80–100 nm). The results showed a clear positive correlation between particle, BC, and BTEX emissions, and a negative correlation between particles and $NO_2$. Additionally, there is a clear correlation between BC emissions and PN or $PM_{10}$ emissions. The study showed a significant decrease in aldehyde emissions compared to other studies such as [238,239]. The aldehyde emissions from the gasoline vehicles decreased from 12 mg $km^{-1}$ for the Euro 1 vehicles to 1 mg $km^{-1}$ for the Euro 4 vehicles. The aldehyde emissions from the diesel vehicles decreased from 30 mg $km^{-1}$ for the pre-Euro vehicles to 0.5 mg $km^{-1}$ for the Euro 4 vehicles. A significant decrease in PAH emissions is observed compared to [239,240]. PAH diesel emissions decreased from 17 μg $km^{-1}$ for pre-Euro vehicles to 1 μg $km^{-1}$ for Euro 3 vehicles and remained below 0.2 μg $km^{-1}$ for Euro 5 vehicles. PAH gasoline emissions were quite low (below 8 ng $km^{-1}$) and close to the detection limit.

To investigate the effects of after-treatment systems on emission of pollutants, a recent study [241] characterised the chemical composition of particles emitted from three diesel and four gasoline Euro 5 light-duty vehicles on a chassis dynamometer facility. BC was the dominant emitted species with emission factors (EFs) varying from 0.2 to 7.1 mg $km^{-1}$ for gasoline cars and 0.003 to 0.08 mg $km^{-1}$ for diesel cars. For gasoline cars, the OM EFs varied from 5 to 103 μg $km^{-1}$ for direct-injection (GDI) vehicles, and from 1 to 8 μg $km^{-1}$ for port fuel injection (PFI) vehicles, while for the diesel cars it ranged between 0.15 and 65 μg $km^{-1}$. PAH emissions from the GDI technology were 4-fold higher compared to the vehicles equipped with a PFI system during the cold start cycle, while the nitro-PAHs fraction was 25-fold much more appreciable in the GDI emissions. For two of the three diesel vehicles, the PAH emissions were close to the detection limit, but for one, which presented an after-treatment device failure, the average PAH EF was 2.04 μg $km^{-1}$. Emissions of nanoparticles (below 30 nm), mainly composed by ammonium bisulphate, were measured during the passive regeneration of the catalysed diesel particulate filter (CDPF) vehicle. Transmission electronic microscopy (TEM) images confirmed the presence of ubiquitous nanometric metal inclusions into soot particles emitted from the diesel vehicle equipped with a fuel-borne catalyst-diesel 30 particulate filter (FBC-DPF). Their findings show that different after-treatment technologies have an important effect on the level and the chemical composition of the emitted particles. In addition, this research highlights the importance of particle filter device condition and regular checking. In line with these analyses, the evolution of toxicity from Euro 3 to Euro 6 diesel particles was studied by Zerboni et al. [242], and diesel exhaust particles from Euro 6 show less PAH content than Euro 3 but with a higher content of some metals (e.g., Fe and Zr).

As mentioned in Section 3.5, IVOCs can be an important precursor of SOA. A recent study [243] characterised I/SVOC emissions using thermal desorption two-dimensional gas chromatography–mass spectrometry with electron impact (GC × GC–EI–MS) and vacuum ultraviolet (GC × GC–VUV–MS) ionisation. They combined new emissions data with published SOA yield parametrisations to estimate SOA formation potential. After 24 h of oxidation, IVOC emissions contributed 45% of SOA formation; BTEX compounds (benzene, toluene, xylenes, and ethylbenzene), 40%; other VOC aromatics, 15%. The composition of IVOC emissions was consistent across the test fleet, suggesting that future reductions in vehicular emissions will continue to reduce SOA formation and ambient particulate mass levels.

Results from smog chamber investigations are available in the literature, characterising the POA and the corresponding SOA formation at atmospherically relevant concentrations for various diesel vehicles with different exhaust after-treatment systems as in Chirico et al. [244]. For the conditions explored in [244], primary aerosols from vehicles without a particulate filter consisted mainly of BC with a low fraction of OM (OM/BC < 0.5), while the subsequent aging by photooxidation resulted in a consistent production of SOA only for the vehicles without a DOC and with a deactivated DOC. The efficiency of the DOC+DPF is clearly highlighted to reduce primary carbonaceous species and SOA (Figure 12); however, when the DPF is not activated, there are some differences in emissions according to the regime studied.

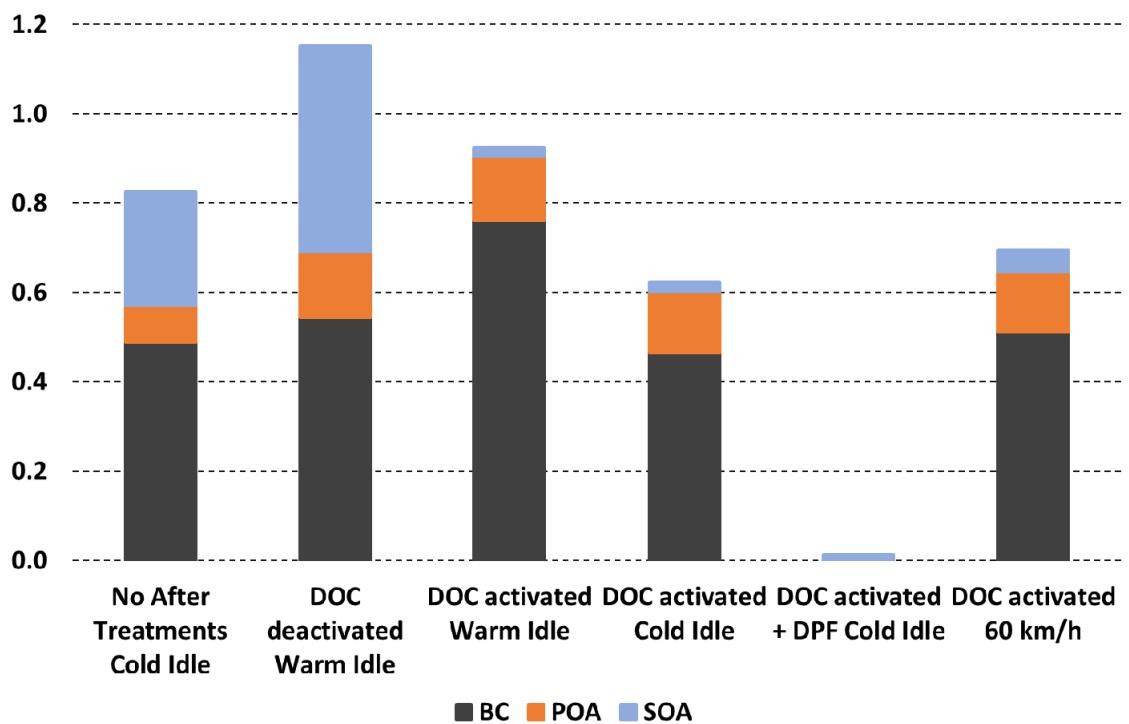

**Figure 12.** Emission factors of POA, SOA and BC in g $kg_{fuel}^{-1}$ from in-use vehicles. SOA produced after 5 h aging. Diagram reproduced from [244].

The number size distributions of particles between 1.2 and 1000 nm were also investigated by Zeraati-Rezaei et al. [245]. Exhaust gas samples were taken before a diesel oxidation catalyst (DOC), after the DOC and after the DOC combined with a catalysed diesel particulate filter (DPF). In samples taken before the DOC (engine-out), the total particulate IVOC+SVOC (I+SVOC) emission factor was approximately 105 mg $kg^{-1}$ of fuel consumed (which was ~49% of the total particle mass) and the peak concentration of different classes of I+SVOC was found in the particle size bins close to 100 nm, where most of the total particle mass was found. Alkanes were the most dominant class of I+SVOC in samples taken before and after the after-treatment devices. Total particulate I+SVOC emissions were removed with ~75% efficiency using the DOC and by ~92% using the DOC+DPF. Alkanes, cycloalkanes, bicyclics and monoaromatics were all removed by >90% using the DOC+DPF; however, oxygenates were removed by only ~76% presumably due to the oxidation of different species within the after-treatment system and reappearance as oxygenates. A high concentration of particles was measured in the sub-2.5 nm range. These particles were efficiently removed by the DOC+DPF due to both the loss of I/SVOCs and physical filtration. This study confirms the high efficiency of after-treatment systems to remove ultrafine particles.

*4.6. The Increasing Role of Gasoline Vehicle*

A systematic examination of carbonaceous PM emissions and parameterisation of SOA formation from modern diesel and gasoline cars at different temperatures (22, −7 °C) during controlled laboratory experiments was reported in a recent study [246]. Carbonaceous PM emissions and SOA formation are markedly higher from gasoline than diesel particle filter (DPF) and catalyst-equipped diesel cars, more so at −7 °C, contrasting with nitrogen oxides (NOx). Higher SOA formation from gasoline cars and primary emission reductions for diesels implies gasoline cars will increasingly dominate vehicular total carbonaceous PM, though older non-DPF-equipped diesels will continue to dominate the primary fraction for some time. Supported by state-of-the-art source apportionment of ambient fossil fuel-derived PM, their results show that whether gasoline or diesel cars are more polluting depends on the pollutant in question, i.e., that diesel cars are not necessarily worse polluters than gasoline cars.

The carbon number can be considered as a picture of the composition of the OA [247] from gasoline and diesel vehicles. Fuel composition, engine design/technology (including computerised control), and exhaust after-treatment technology are all determining factors in the composition of emissions. Gasoline engines use spark ignition of a lower-molecular-weight fuel with compounds in the C4−C10 range (premixed with air) compared to diesel engines, with compression ignition of heavier fuels (C9−C25) at higher temperatures and fuel-lean conditions to achieve greater efficiencies. Reformulations have reduced volatility, reactivity, trace impurities, and/or propensity for forming NOx, CO, or BC in the exhaust.

The co-formation and co-release of genotoxic PAHs, alkyl-PAHs and soot nanoparticles from gasoline direct-injection vehicles have been investigated by Muñoz et al. [248]. In their study, they hypothesised that particles are released together with PAHs formed under the same combustion conditions. They showed that particle emissions increased by 2–3 fold during acceleration like CO, indicating that transient driving produces fuel-rich conditions with intense particle formation. For comparison, an Euro 5 diesel vehicle (1.6 L) equipped with a particle filter released $3.9 \times 10^{10}$ particles km$^{-1}$ (cWLTC), clearly within the Euro 5/6 limit value of $6.0 \times 10^{11}$ particles km$^{-1}$ and 64-fold below the GDI fleet average. Mean PAH emissions of the GDI fleet were 2-fold higher than the benchmark diesel vehicle. A comparison of the toxicity equivalent concentrations in the cWLTC of the GDI fleet and the diesel vehicle revealed that GDI vehicles released 6–40-fold higher genotoxic PAHs than the diesel vehicle. PAHs adsorbed on numerous soot nanoparticles are introduced into the lung by the property of sub-200 nm particles being deposited in the alveoli. They showed that all GDI vehicles tested released large numbers of nanoparticles carrying substantial loads of genotoxic PAHs. If non-treated diesel exhaust is considered a class-1 carcinogen by the WHO, inducing lung cancer in humans, these GDI vehicle exhausts may be a major health risk too for those exposed to them, corroborating the progress achieved with current diesel vehicles, now equipped with efficient particle filters.

The BC emissions were categorised from passenger cars in South Korea [249] according to fuel type and the vehicles were categorised by the engine technology used and the standards applied. In all types of vehicles, BC emission was reduced with driving speeds from low to high, but beyond a certain threshold, the emission increased again. Diesel vehicles with DPF and liquefied petroleum gas (LPG) vehicles showed low levels of emissions at 0.01 mg km$^{-1}$ or less in certain modes, and at 0.045 and 0.0009 mg km$^{-1}$ in regulation modes, respectively. Among gasoline vehicles, GDI vehicles showed BC emissions of 2.8–5-fold higher than that of multi-point injection (MPI) vehicles. With the tightening of regulations, MPI vehicles showed reduced PM emissions other than BC. Therefore, with the application and tightening of PM regulations, both gasoline and diesel vehicles exhibited reduced BC emissions.

For various types of vehicles, gasoline DI vehicles emitted 25% more $CO_2$ than diesel vehicles for all ARTEMIS and NEDC driving conditions, as mentioned in a recent study [250]. The GDI emitted 2- to 200-fold more PN and BC and 5- to 150-fold less NOx than diesel vehicles. Comparing to diesel catalysed DPF, additive DPF vehicles

emitted 2-fold more $NO_2$. No significant differences were observed between additive and catalysed DPF for $CO_2$ and NOx emissions. Moreover, a clear impact of cold start on emission was observed during their experiments. BC emissions were the highest by far for the GDI reaching $1531 \pm 207 \, \mu g \, km^{-1}$ and exhibited the highest PN emissions for the cold start ARTEMIS urban cycle.

Regarding IVOC emissions from gasoline vehicles leading to SOA formation, along with the stricter emission standards, VOC emissions could be the most effectively controlled organic gases according to a recent study [157]. While the mitigation efficiency of organic VOCs decreased as the emission standards improved, this indicates the growing importance of organic VOC mitigation for more restrictive emission standards vehicles. In addition, the relative contributions of IVOCs and unidentified NMHCs are becoming increasingly prominent in total organic gas emissions of gasoline vehicles.

### 4.7. Impact of Alternative Fuels on Emissions

The potential of renewable fuel to reduce diesel exhaust particle emissions has been recently tackled [251]. Diesel exhaust emissions including particle number concentration and size distribution along with the particles' chemical composition and NOx were investigated from a Euro 4 passenger car with a comprehensive set of high time-resolution instruments. The emissions were compared with three fuel standards—European diesel (EN590), Indian diesel (BS IV) and Finnish renewable diesel (Neste MY)—over the New European Driving Cycle (NEDC) and the Worldwide harmonised Light vehicles Test Cycle (WLTC). Fuel properties and driving conditions strongly affected exhaust emissions. The exhaust particulate mass emissions for all fuels consisted of BC (81–88%), with some contribution from organics (11–18%) and sulphate (0–3%). As aromatic-free fuel, the MY diesel produced approximately 20% lower BC emissions compared to the EN590 and 29–40% lower compared to the BS IV. High volatile nanoparticle concentrations at high WLTC speed conditions were observed with the BS IV and EN590 diesel, but not with the sulphur-free MY diesel. These nanoparticles were linked to sulphur-driven nucleation of new particles in cooling dilution of the exhaust [252]. For all the fuels, non-volatile nanoparticles in sub-10 nm particle sizes were observed during engine braking, and they were most likely formed from lubricant-oil-originated compounds [253]. With all the fuels, the measured particulate and NOx emissions were significantly higher during the WLTC cycle compared to the NEDC cycle. This recent study demonstrated that renewable diesel fuels enable mitigations of particulate and climate-warming BC emissions of traffic and would simultaneously help tackle urban air quality problems.

Hydrotreated vegetable oil (HVO) diesel fuel was considered a promising biofuel candidate that could complement or substitute traditional diesel fuel in engines [254,255]. It has been already reported that changing the fuel from conventional EN590 diesel to hydrotreated vegetables oils (HVO) decreases exhaust emissions up to −30% for PM. However, as the fuels have certain chemical and physical differences, it is clear that the full advantage of HVO cannot be realised unless the engine is optimised for the new fuel. In this article, they studied how much exhaust emissions can be reduced by adjusting engine parameters for HVO. The results indicate that, with all the studied loads (50%, 75%, and 100%), particulate mass and NOx can both be reduced over 25% by engine parameter adjustments. Further, the emission reduction was even higher when the target for adjusting engine parameters was to exclusively reduce either particulates or NOx. The use of HVO is also expected to reduce SOA formation compared to a more standard fuel, as shown by Ghadimi et al. [256].

The effect of biofuel use on the operation of the diesel exhaust after-treatment System was investigated both numerically and experimentally in a dedicated study [257] by focusing on the contribution of three main factors: raw PM–NOx emissions trade-off, the NO–$NO_2$ conversion efficiency of the diesel oxidation catalyst (DOC) and PM reactivity toward oxidation. The experimental results indicate a significant reduction in soot emissions, in line with literature trends, especially at high loads, as fuel oxygen enhances oxidation in

the fuel-rich regions of the combustion chamber. On the other hand, only a slight increase in NOx emissions has been observed, along with a similar trend in the equivalence ratio due to both the lower heating value and stoichiometric air/fuel ratio of biodiesel in comparison with fossil fuelling.

Other results [258] indicated that the use of SCR and the largest fraction of biodiesel studied may suppress the emission of total PAHs. The toxic equivalent (TEQ) was lower when using 20% biodiesel, in comparison with 5% biodiesel on the SCR system, reaffirming the low toxicity emission using a higher-percentage biodiesel. The data also reveal that use of SCR, on its own, suppresses nitro-PAH compounds. In general, the use of larger fractions of biodiesel (B20) coupled with the SCR after-treatment showed the lowest PAH and nitro-PAH emissions, meaning lower toxicity and, consequently, a potential lower risk to human health. From an emissions point of view, the results of the study also demonstrated the viability of the biodiesel program, in combination with the SCR systems, which does not require any engine adaptation and is an economical alternative for countries that have not adopted Euro VI emission standards.

The use of synthetic fuels combined with a dual-mode dual-fuel (DMDF) combustion strategy could be a potential combustion mode to achieve ultra-low soot emissions, as requested by the future regulations on internal combustion engines [259]. In the case of DMDF, poly-oxymethylene dimethyl ether (OMEx) has been found as a promising alternative to diesel when combined to gasoline. In this sense, the OMEx–gasoline combination promotes zero-soot emissions while maintaining engine performance and acceptable levels of other regulated emissions.

Similarly, the effect of diesel fuel blended with polyoxymethylene dimethyl ethers (PODE$_n$) as a new alternative fuel was studied to meet the China VI emission standards under various testing conditions [260]. According to the authors, blending PODEn into diesel could effectively improve the emission characteristics of diesel engines. In high-speed and high-load conditions, the PODEn, due to its high oxygen content and great volatility, improved the oxygen concentration in the cylinder and led to better uniformity of the combustible mixture, which effectively inhibited the generation of soot by up to 76%. With the increase in the PODEn blending ratio, NOx emission increased slightly, but emissions of HC and CO gradually decreased, with the highest reduction in CO emissions reaching approximately 66%.

Lubrificating oils is also a source of SVOC emissions, as stated by Liang et al. [261]. An analysis based on PMF illustrated that diesel fuel contributed to the SVOC emissions by nearly 70% at low load (442 µg kg fuel$^{-1}$), while lubricating oil dominated the SVOC emissions by over 50% at high load (595 µg kg fuel$^{-1}$). They also found that combustion by-products had non-negligible contributions to the SVOC emissions, increasing from 13% to 38% as load increased.

*4.8. Impact of Real Driving Conditions*

The characterisation of real-drive emissions from three Euro 6 emission-level passenger cars (two gasoline and one diesel) in terms of fresh particles and secondary aerosol formation has been assessed [262]. The gasoline vehicles were also characterised by chassis dynamometer studies. In the real-drive study, the particle number emissions during regular driving were 1.1–12.7-fold greater than observed in the laboratory tests (4.8-fold greater on average), which may be caused by a more effective nucleation process when diluted by real polluted and humid ambient air. However, the emission factors measured in the laboratory were still much higher than the regulatory value of $6 \times 10^{11}$ particles km$^{-1}$. The higher emission factors measured here probably result from the fact that the regulatory limit considers only non-volatile particles larger than 23 nm, whereas, here, all particles (also volatile) larger than 3 nm were measured. Secondary aerosol formation potential was the highest after a vehicle cold start when most of the secondary mass was organics. After the cold start, the relative contributions of ammonium, sulphate and nitrate increased. Using a novel approach to study secondary aerosol formation under real-drive conditions

with the chase method resulted mostly in emission factors below detection limit, which was not in disagreement with the laboratory findings. For all of the vehicle types in a study focusing on Euro 5 and 6 vehicles [263], solid PN emissions increased with lower ambient temperatures. While for GDI and PFI, the sub-23 nm fraction decreased at low temperatures, it increased for DPF vehicles.

Geldenhuys et al. [264] showed that PAH profiles for each operating mode differed significantly in terms of the number, type and concentration of PAHs. Total PAH EFs (gas and particle phase) were determined to be 1181.14 and 592.10 µg kg$^{-1}$ for the idle and maximum power mode, respectively. Naphthalene was found to be the most abundant PAHs in the raw exhaust stream for both modes, a total of 9 PAHs were detected in the idle mode and only 4 in maximum power mode. The maximum power mode revealed the highest concentration of particulate PAHs, which correlated with elevated soot measurements.

Regarding the emission of a EUROVI HDV in real driving conditions, Grigoratos et al. [265] found that solid PN emission levels of tested vehicles do not appear to be of concern due to the effectiveness of the currently available DPF systems. Calculated emission factors are one order of magnitude lower than the current laboratory type approval limit and appear to be at the lower limit of the range given in the literature for older-technology HDVs featuring a DPF system. In real-world conditions, the evolution of BC concentrations has been studied [266] for a tunnel in San Francisco; compared to baseline measurements made in 2010 at the same location, the median truck model year observed in 2018 increased by 9 years, and DPF and SCR penetration increased from 15 to 91% and 2 to 59%, respectively. Over this period, fleet-average emission rates of BC and NOx decreased by 79 and 57%, respectively.

### 4.9. Reflexions on Post Euro 6 Regulations

The European Commission has started the regulatory work to define the next emission standards [44]. Post-Euro 6 standards are expected to continue to improve the emission performance of new road vehicles, addressing their contribution to the persistent air quality issues across Europe. To assist in the formulation of these Euro 7 and Euro VII proposals, the Commission have contracted members of Consortium for Ultra-Low Vehicle Emissions (CLOVE) [267]. The International Council on Clean Transportation (ICCT) made some recommendations, summarised in Table 7, on the different topics that should be considered for the light-duty post-Euro 6 standards.

**Table 7.** Summary of recommendations regarding PM carbonaceous species and UFP for post-Euro 6 standards from ICCT [44].

| | |
|---|---|
| Limits | -Introduce fuel- and technology-neutral emission limits<br>-Tighten the emission limits to harmonize with other markets<br>-Introduce application-neutral emission limits |
| Ultrafine particles | -Lower the size cut-off for particle counting from 23 nm to at least 10 nm<br>-Develop a methodology to measure volatile and semi-volatile particles<br>-Include emissions that occur during filter regeneration<br>-Make particulate number (PN) standards fuel and technology neutral<br>-Investigate the feasibility of PN tailpipe measurements |
| Unregulated pollutants | -Set limits for aldehyde emissions<br>-Regulate all VOCs and not just HC<br>-Set emission limits for brake wear particles |

These recommendations on PN and semi-volatile are supported by international research such as recent work [268] on gasoline vehicles gasoline direct injection (GDI) in Beijing. Their results indicated that GDI-engine vehicles emitted a large amount of both primary and secondary OA. PM number emissions of organic particles from GDI-engine vehicle were $2.9 \times 10^9$ particles per kg fuel during the Beijing Driving Cycle. Secondary organic particles were predominant in the secondary aerosols, accounting for 80–85% of

particles in the chamber. Their results also showed that POA emitted by GDI-engine vehicles could acquire OA and sulphate coatings rapidly, within a few hours, and increase a sizable fraction of total ambient aerosols existing as internal mixtures. In addition, the fast ageing further caused the increase in aged POA in the total OA; this consequently largely modified the properties of the particles such as their optical properties. The results of the experiments in the chamber showed that most of the aged POA had a core–shell structure, whereas most of the SOA produced by gas-phase reactions had a uniform structure. In terms of particle size, the particles exhibited a bimodal distribution in number vs. size, with one mode at 800–900 nm and the other at 140–240 nm.

In other research [269], the measured PN from all recent diesel Euro 6 vehicles of the study also point to the very good performance of DPFs and DOCs during real-world operation. However, pollutant emissions that were not a matter of concern at the time the Euro 6 standards were developed, such as CO emissions from gasoline vehicles, or PN emissions from port fuel injection (PFI) gasoline vehicles, were shown to be very high in some instances. GDI vehicles exhibited high PN emissions but PN emissions, sometimes $>6 \times 10^{11}$ # $km^{-1}$, were also recorded for PFI. The only GPF-equipped vehicle exhibited the lowest PN emissions of the gasoline vehicles by far (more than one order of magnitude below the PN limit).

Results studying on-road emissions of Euro 6d-TEMP passenger cars on Alpine routes during the winter period [270] added to the body of evidence indicating that, following the introduction of the RDE in the EU, more efficient emission control technologies are being used to reduce emissions of NOx and PN—the pollutants covered by this test procedure—particularly NOx from diesel vehicles. However, emissions of PN from the two gasoline vehicles were high ($>1 \times 10^{12}$ # $km^{-1}$) under certain experimental conditions. This result is a reason for concern, because PN emissions from vehicles using the fuel injection technology (PFI, port fuel injection) investigated in this study are not limited by the current Euro 6 regulation. This underlines the need for technology- and fuel-neutral approaches to vehicle emission standards, whereby all vehicles must comply with the same emission limits for the same pollutants regardless of the technologies or fuels employed.

A recent study showed that for electric vehicle (EV) switch scenarios, $PM_{2.5}$ emissions could remain unchanged due to the primary contribution of non-exhaust emissions, suggesting that EVs are likely to yield smaller changes in exposure to $PM_{2.5}$ than for NOx. This would be in line with some options planned by the European Commission in its roadmap to revise the Euro6/VI standards by setting stringent thresholds of ELV and address new pollutants and sources with a possible Euro 7/VII regulation entering in force in 2025 [271]. However, since the fraction of EC in PM is smaller for non-exhaust emissions, the switch to EV could be more efficient in reducing carbonaceous emissions of vehicles. Trapping brake wear particles in the braking system before release into ambient air and reducing the material that is tracked onto public road surfaces as a result of vehicle movements in and out of construction sites, waste-management sites, quarries, farms, are cited as mitigation strategies [272].

Regarding ultrafine particles, a recent study in Japan by Fugitani et al. [273] showed that, although PN are not directly regulated in most parts of the world, the effect of PM mass regulation on the PN is of interest. Their study reveals that the policy of regulating emissions in terms of PM mass did not directly reduce PN, but still did so to some extent, as of 2016. They concluded that control policies for condensable particles and their precursors are needed to further reduce the vehicle-exhaust-derived PNs in the environment. The 23 nm lower size does not always cover the whole or the largest part of the size distribution of the emitted particles, as highlighted in a recent review [274] and references therein. High amounts of sub-23 nm particles are measured very often, both for diesel and gasoline direct-injection vehicles, but in particular for the technologies not subject currently to a SPN limit (port fuel injection gasoline and gas engines). On the other hand, for heavy-duty vehicles, all technologies are subject to the same limit. For this reason, it has been suggested to both (i) lower the diameter from 23 to 10 nm and (ii) include all technologies in the limits

without exceptions. This statement is supported by Selleri et al. [275] with an increase in PN by 25% if one includes particles between 10 and 23 nm.

## 5. Conclusions and Recommendations

From this exhaustive literature review focussed on carbonaceous PM, PAHs and ultrafine particles, the following statements can be drawn to illuminate the decision-making process on air pollution regulations currently in progress in Europe and worldwide regarding vehicle emissions. Recommendations are highlighted in bold characters.

- The residential sector (heating systems) and biomass burning remain the largest carbonaceous PM emitters at the global scale. Vehicle emissions remain an important contributor (the 2nd in Europe) to the carbonaceous fraction of PM, particularly in urban areas. BC, PAHs and UFP concentrations have an impact on human health and climate.

- PM produced by combustion emitted at the exhaust pipe are mostly fine particles below 2.5 µm and are mainly composed of carbonaceous species. Carbonaceous species (BC, OC and BrC) dominate the composition of PM issued from the exhaust pipe. Emitted IVOCs and SVOCs, if not condensing directly as particulate POA, can react in the atmosphere to form particulate SOA.

- **We must understand what type of organic gases are in the usual HC or VOCs category in emission inventories. We recommend for modelling applications to specify the all carbonaceous emissions by volatility bins (from very low to highly volatile species).**

- There is a consensus a reduction in carbon-containing vehicular emissions is beneficial for both air quality and climate change issues.

- However, it remains unclear whether BrC is underestimated in emission factor and a there is an urgent need to disentangle BrC from BC and/or OC because the optical properties of BrC are closest to that of BC. If BrC is supposed to be in the OC fraction in climate models, the warming effect of aerosols should be underestimated. **A clear estimate of the BrC fraction in emission inventories would be desirable**.

- PM, BC, PN, and PAH emissions are effectively reduced using tailpipe after-treatment systems as diesel particulate matter (DPF) or gasoline particulate matter (GPF). Decreases from 90 to 100% are commonly observed for most particulate pollutants.

- An overall reduction in PM mass could theoretically enhance the formation of ultrafine particles and then increase the PN through the nucleation of organic and inorganic species because of a reduction in absorbing material. However, these new UFP may have a short lifetime. All these phenomena remain unclear. **There is a need to better understand the process of dilution of exhaust emissions from the tailpipe to ambient conditions with modelling tools and field observations, but the latest results show an interest in measuring PN up to 10 nm.**

- Since species emitted at the exhaust can quickly evolve during dilution, it could be relevant to account for aging processes to calculate a more relevant EF to be used in models. **There is certainly an issue to consider in terms of characteristic time/length after ejection in line with the resolution of the air quality model resolution, but the formulation of any such EFs should be performed as a cooperation between the emission and modelling communities.**

- Successive Euro regulations have substantially reduced emissions of POA, including condensables. After-treatment systems reduce the intermediate-volatility organic compounds (IVOC) emissions but there are still several gaps in the knowledge of these compounds and their chemical transformation after emissions and in ambient conditions.

- With the decrease in PM emissions, even recent gasoline vehicles can now produce more particle numbers. The use of GPF for gasoline is a key technology to reduce PN and PM emissions. However, a study has reported larger genotoxic PAH emissions from gasoline vehicles (mainly gasoline direct-injection vehicles) even equipped with

DPF compared to diesel equipped with DPF (2-fold higher). **PAH emissions must be monitored more with an increasing share of gasoline vehicles.**

- Recent research findings show that different after-treatment technologies have an important effect on the level and the chemical composition of the emitted particles, and highlight the importance of particle filter device conditions and regular checking to maintain best performance. **Maintenance of after-treatment systems is key to maintaining best performance.**

- The role of real driving emissions is clearly identified, with quite different emissions factors largely depending on engine operating mode. **Particular attention should be paid to pursue the use of real-world driving cycles to determine EF.**

- For non-equipped diesel vehicles, the use of biofuels can reduce BC emissions by 30% and could be an option to achieve the legal air quality target or limit values sooner.

- Even if brake, tyre and road wear emit mainly coarse particles, a non-negligible fine fraction of PM is emitted. The TSP emissions per km are larger than current Euro VI emissions and a similar fraction of BC is observed in exhaust and no-exhaust PM emissions. Brakes also produce ultrafine particles, metals and PAHs, and temperature greatly affects dust PM emissions. BC emissions from brakes are correlated to $PM_1$ emissions.

- Under particular driving conditions, UFP below 60 nm can be emitted during braking phases due to tyre emissions; but compared to DPF-equipped cars, emissions remain of minor importance.

- There is no widely used after-treatment system to control brake, tyre and road wear emissions. The type of materials and the behaviour of the driver are often cited as key to reducing emissions. Some companies have developed brake particle collection systems that would reduce brake mass and number emissions by 80% to 90%, respectively.

- **For particle number measurements, it is important to define the methodology details (sampling and dilution, measurement instrumentation, relevant sizes, etc.) and conduct inter-laboratory exercises to determine the reproducibility of a proposed method.**

- **PM resuspension from the road should be addressed and analysed in terms of carbonaceous composition avoiding double counting in emission inventories**. In principle, these types of emissions are not yet regulated, although they are responsible for a large fraction of total road traffic emissions. Such emissions depend on meteorology (wind, temperature, humidity, precipitation) and site climatology/location (land use in the vicinity).

Ultrafine particles containing carbon are known to have adverse effects on health. Even if the solid fraction (BC/EC) of a particle has been reduced by after-treatment systems, the processes involving secondary species formation are still poorly known and this organic matter dominates the carbonaceous particulate emission of vehicles. A topic still poorly addressed so far in this research field is the role of physicochemical interactions between organic and inorganic species during the dilution process of vehicular emissions. From a research perspective, we must emphasise the need to better understand the processes at work during the dilution phase of the emissions with both adequate measurement systems and modelling tools. The evolution of the mixing state of emitted particles is also an important issue in terms of their impact on health as well as on radiative impacts.

**Author Contributions:** Conceptualisation, B.B.; methodology, B.B.; investigation, B.B.; writing—original draft preparation, B.B.; data analysis, B.B. and J.-P.P.; writing—review and editing, N.A., J.-P.P., F.C., J.-M.A., P.T., E.P., D.S. and B.N.M.; visualisation, B.B.; supervision, B.B.; project administration, B.B.; All authors have read and agreed to the published version of the manuscript.

**Funding:** This research received no external funding.

**Institutional Review Board Statement:** Disclaimer—The U.S. Environmental Protection Agency (US EPA) through its Office of Research and Development collaborated in the research described here. The views expressed in this article are those of the authors and do not necessarily represent the views or policies of US EPA. Any mention of trade names, manufacturers or products does not imply an endorsement by the United States Government or the U.S. Environmental Protection Agency. EPA and its employees do not endorse any commercial products, services, or enterprises.

**Informed Consent Statement:** Not applicable.

**Data Availability Statement:** Not applicable.

**Acknowledgments:** The authors acknowledge Leonidas Ntziachristos (Aristotle University of Thessaloniki) for fruitful discussions on HC analysis techniques.

**Conflicts of Interest:** The authors declare no conflict of interest.

## List of Acronyms and Abbreviations

| | |
|---|---|
| AAE | Absorption Ångström Exponent |
| AC | Arctic Council |
| ACAP | Arctic Contaminants Action Program |
| AGP | Amended Gothenburg Protocol |
| AMS | Aerosol Mass Spectrometer |
| APG | Associated Petroleum Gas |
| AQ | Air Quality |
| ARTEMIS | Assessment and Reliability of Transport Emission Models and Inventory Systems |
| BaP | Benzo(a)pyrene |
| BB | Biomass Burning |
| BC | Black Carbon |
| BID | Barrier discharge Ionisation Detector |
| BrC | Brown Carbon |
| BTEX | Benzene, Toluene, Ethylbenzene, Xylenes |
| CC | Climate Change |
| CCAC | Climate and Clean-Air Coalition |
| CDPF | Catalysed Diesel Particulate Filter |
| CEN | Comité Européen de Normalisation—European Committee for Standardisation in English |
| CEIP | Centre on Emission Inventories and Projections |
| CLOVE | Consortium for Ultra-Low Vehicle Emissions |
| CLRTAP | Convention on Long-Range Transboundary Air Pollution |
| CO | Carbon Monoxide |
| CO2 | Carbon Dioxide |
| DI | Diesel |
| DOC | Diesel Oxidation Catalyst |
| DPF | Diesel Particulate Filter |
| DRF | Direct Radiative Forcing |
| DT | Dilution Tunnel |
| EB | Executive Body |
| eBC | Equivalent BC |
| EC | European Commission or Elemental Carbon |
| ECE | Economic Commission for Europe |
| ECLIPSE | Evaluating the Climate and air quality ImPacts of Short-livEd Pollutants |
| EECCA | Eastern Europe, Caucasus and Central Asia |
| EEA | European Environmental Agency |
| EF | Emission Factor |
| EGR | Exhaust Gas Recirculation |
| ELV | Emission Limit Values |
| EMEP | European Monitoring and Evaluation Programme |
| EU | European Union |

| | |
|---|---|
| FID | Flame Ionisation Detection |
| FP | Fine Particle |
| FTP | Federal Test Procedure |
| GAINS | Greenhouse Gas–Air pollution Interactions and Synergies (IIASA model) |
| GC–MS | Gas Chromatography–Mass Spectrometry |
| GDI | Gasoline Direct Injection |
| GFED | Global Fire Emissions Database |
| GHG | Greenhouse Gas |
| GPF | Gasoline Particle Filter |
| HC | Hydrocarbons |
| HDV | Heavy-Duty Vehicle |
| HOM | Highly Oxidised Matter |
| HRTEM | High-Resolution Transmission Electron Microscopy |
| HULIS | Humic-Like Substance |
| HVO | Hydrotreated Vegetable Oil |
| ICCT | International Council on Clean Transportation |
| IVOCs | Intermediary Volatility Organic Compounds |
| IR | InfraRed |
| JRC | Joint Research Centre |
| LCV | Light Commercial Vehicle |
| LDV | Light-Duty Vehicle |
| LEV | Low Emission Vehicles |
| LLE | Loss of Life Expectancy |
| LPG | Liquefied Petroleum Gas |
| MCE | Modified Combustion Efficiency |
| MPI | Multi-Point Injection |
| NDC | Nationally Determined Contributions |
| NEDC | New European Driving Cycle |
| NEE | Non-Exhaust Emissions |
| NFR | Nomenclature For Reporting |
| NMVOCs | Non-Methane Volatile Organic Compounds |
| NOx | Nitrogen Oxides |
| NP | Nanoparticles |
| OC | Organic Carbon |
| OM | Organic Matter |
| PAHs | Polycyclic Aromatic Hydrocarbons |
| PAM | Potential Aerosol Mass |
| PC | Passenger Car |
| PFI | Port Fuel Injection |
| PM | Particulate Matter |
| $PM_x$ | Particulate Matter for particle diameter below x μm (x = 0.1, 1, 2.5, 10) |
| PN | Particle Number |
| POA | Primary Organic Aerosols |
| PODE | Polyoxymethylene Dimethyl Ethers |
| POP | Persistent Organic Pollutant |
| PUF | Polyurethane Foam |
| RDE | Real Driving Emissions |
| rBC | Refractory BC |
| SCR | Selective Catalyst Reduction |
| SDG | Sustainable Development Goal |
| SLCP | Short-Lived Climate Pollutants |
| SMPS | Scanning Mobility Particle Sizer |
| SNAP | Selected Nomenclature for Air Pollution |
| SOA | Secondary Organic Aerosol |
| SOx | Sulphur Oxides |
| SVOCs | Semi-Volatile Organic Compounds |

| | |
|---|---|
| TEM | Transmission Electron Microscopy |
| TFIAM | Task Force on Integrated Assessment Modelling |
| TFTEI | Task Force on Techno-Economic Issues |
| THC | Total HC |
| TSP | Total Suspended Particles |
| TWC | Three Way Catalyst |
| UCM | Unresolved Complex Mixture |
| UFP | Ultrafine Particle |
| ULEV | Ultra Low Emission Vehicle |
| UN | United Nations |
| UNECE | United Nations Economic Commission for Europe |
| UNFCCC | United Nations Framework Convention on Climate Change |
| U.S. | United States |
| UV | Ultraviolet |
| VOCs | Volatile Organic Compounds |
| WHO | World Health Organisation |
| WMO | World Meteorological Organisation |
| WLTC | Worldwide harmonised Light vehicles Test Cycle |

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
