# Peer review of "Emissions of Carbonaceous Particulate Matter and Ultrafine Particles from Vehicles—A Scientific Review in a Cross-Cutting Context of Air Pollution and Climate Change"

_applsci, doi:10.3390/app12073623_

Round 1

Reviewer 1 Report

Abstract section should be revised completely. This review paper is dedicated to the

Emissions of Carbonaceous Particulate Matter and Ultra Fine Particles from Vehicles. So abstract would be in a broader perspective rather than for specefic region. You can add

Europe in the discussion part or as a case study.

  1. It would be better to introduce a table describing different BC PM, and

other relevant information.

  1. Author should strictly follow the proper format of MDPI.
  2. The similarity index of your Paper is 28%. Author should revise the whole chapter

and reduce similarity index up to 18% max.

  1. Please cross check if references are properly given within text. Check for appropriate formatting.
  2. Cross-check the paper for grammatical errors.
  3. Replace “PM10” and “PM2.5” with “PM10” and “PM5” throughout the manuscript.
  4. Page No. 5: Line No., 202 – It is mentioned that wildfires emissions are major drivers of primary PM emissions. Please cross-check this line.
  5. Page No. 8: Line No., 282 to 284 – Methane is mentioned as an example of Short-Lived Climate Pollutants. Please clarify as methane has a longer atmospheric lifetime.
  6. Page No. 10: Line No., 373 – Improper use of punctuations.
  7. Page No. 16: Line No., 610 to 622 – It is stated that there are seven categories of POA but only six are mentioned. Please cross-check.
  8. I would strongly recommend that Authors should provide the permission log with details of all figures/tables taken from third party sources and permission copies for the same
  9. Also, Permission from the rightsholder for reusing figures/tables (if any).

Author Response

Response to Reviewer 1 Comments

We would like to thank the reviewer for her/his positive and constructive feedback, which helped to improve the quality of the paper. Below we address specific issues mentioned by the reviewers point by point. The manuscript has been updated accordingly.

Abstract section should be revised completely. This review paper is dedicated to the Emissions of Carbonaceous Particulate Matter and Ultra Fine Particles from Vehicles. So abstract would be in a broader perspective rather than for specific region. You can add Europe in the discussion part or as a case study.

In response to this general comment, we have suppressed the mention of Europe in the abstract. The abstract has been modified.

Point 1. It would be better to introduce a table describing different BC PM, and other relevant information.

Response 1. There are already several diagrams or tables showing the various types of carbon and PM composition Table 1, Table 5, Figure 4 and 6. We are not sure that a new figure/table is necessary.

Point 2. Author should strictly follow the proper format of MDPI.

Response 2. The format of MDPI has been strictly followed, the official template is used

Point 3. The similarity index of your Paper is 28%. Author should revise the whole chapter and reduce similarity index up to 18% max.

Response 3. We have shortened (particularly section 4.1 and 4.2) by removing too detailed statements and revised the text, we hope the similarity index will be then lower. Moreover, we have discussed this point with the editor. Indeed, 40% of this paper is based on a report considered as a non-peer reviewed grey literature. This report is available on internet and would explain this similarity index.

Point 4. Please cross check if references are properly given within text. Check for appropriate formatting.

Response 4. We use Zotero ensuring an adequate automatic referencing with MDPI style. Some corrections have been done in the references.

Point 5. Cross-check the paper for grammatical errors.

Response 5. We have cross-checked the paper

Point 6. Replace “PM10” and “PM2.5” with “PM10” and “PM2.5” throughout the manuscript.

Response 6. There is no consensus on the use of indexes or not as PM is not a molecule. However, we have made the modification and also for PM1 and PM0.1.

Point 7. Page No. 5: Line No., 202 – It is mentioned that wildfires emissions are major drivers of primary PM emissions. Please cross-check this line.

Response 7. We agree, it is now corrected.

Point 8. Page No. 8: Line No., 282 to 284 – Methane is mentioned as an example of Short-Lived Climate Pollutants. Please clarify as methane has a longer atmospheric lifetime.

Response 8. The reviewer is right we have removed “Methane” from the discussion.

Point 9. Page No. 10: Line No., 373 – Improper use of punctuations.

Response 9. We agree, it is now corrected

Point 10. Page No. 16: Line No., 610 to 622 – It is stated that there are seven categories of POA but only six are mentioned. Please cross-check.

Response 10. We agree, it is now corrected, indeed it was six.

Point 11. I would strongly recommend that Authors should provide the permission log with details of all figures/tables taken from third party sources and permission copies for the same

Response 11. All figures and tables are original made by the first author with the Microsoft software suite. Sometimes, they are inspired by papers that are properly cited. All figures will be provided in high resolution pdf.

Point 12. Also, Permission from the rightsholder for reusing figures/tables (if any).

Response 12. See previous comment, there is no need to ask permissions in our case

Reviewer 2 Report

This review focuses on carbonaceous PM and gaseous precursors from road traffic emissions, including ultrafine particles (UFP) and polycyclic aromatic hydrocarbons (PAH) that are clearly associated with the evolution and formation of carbonaceous species, a Areas of research of interest, especially discussed in the context of the intersection of air pollution and climate change. This article will be of interest to many readers. Modifications and suggestions are as follows:
 1) This paper is very long, although the author has strengthened the abstract and table of contents, it may be better to do a detailed framework diagram instead of table of contents.
2) At the end of the article, I hope to see the issues that should be focused on and need to be explained in the next step in this field, and the research direction, which is the ultimate purpose of the review.

Author Response

Response to Reviewer 2 Comments

We would like to thank the reviewer for her/his positive and constructive feedback, which helped improve the quality of the paper. Below we address specific issues mentioned by the reviewers point by point. The manuscript has been updated accordingly.

This review focuses on carbonaceous PM and gaseous precursors from road traffic emissions, including ultrafine particles (UFP) and polycyclic aromatic hydrocarbons (PAH) that are clearly associated with the evolution and formation of carbonaceous species, a Areas of research of interest, especially discussed in the context of the intersection of air pollution and climate change. This article will be of interest to many readers. Modifications and suggestions are as follows:

Point 1. This paper is very long, although the author has strengthened the abstract and table of contents, it may be better to do a detailed framework diagram instead of table of contents.

Response 1. We have shortened the abstract and some parts of the paper to gain in readability. We are not convinced that a new diagram will help the reader. We have made an effort to shorten the paper removing some statements the less relevant for the scope of the paper.

Point 2. At the end of the article, I hope to see the issues that should be focused on and need to be explained in the next step in this field, and the research direction, which is the ultimate purpose of the review.

Response 2. The reviewer is right we have added this concluding remark:

Ultra-fine particles containing carbon is known to have adverse effects on health. Even if the solid fraction (BC/EC) of particle has been reduced by after treatment systems, the processes involving secondary species formation is still poorly known and this organic matter dominates the carbonaceous particulate emission of vehicles. A topic still poorly addressed so far in this research field is the role of physico-chemical interactions between organic and inorganic species during the dilution process of vehicular emissions. In a research perspective, we must emphasize the need to better understand the processes at work during the dilution phase of the emissions with both adequate measurement systems and modelling tools. The evolution of the mixing state of emitted particles is also an important issue for their impact on health and radiative impacts.

Reviewer 3 Report

The authors presented a comprehensive review of the particles released into the air during combustion processes. They also included a passage on alternative sources. The review is divided into clear sections, the English language is fine, I have no problem publishing the article in this form.

Author Response

Response to Reviewer 3 Comments

Point 1. The authors presented a comprehensive review of the particles released into the air during combustion processes. They also included a passage on alternative sources. The review is divided into clear sections, the English language is fine, I have no problem publishing the article in this form.

Response 1. We thank the reviewer for her/his time to review the paper and this very positive feedback.

Round 2

Reviewer 1 Report

Authors have made relavant changes and I hope it can be accepted now.